



# Hydrological response to warm and dry weather: do glaciers compensate?

Marit Van Tiel[1], Anne F. Van Loon[2], Jan Seibert[3], and Kerstin Stahl[1]

[1]Environmental Hydrological Systems, Faculty of Environment and Natural Resources, University of Freiburg, Germany
[2]Institute for Environmental Studies, Vrije Universiteit Amsterdam, the Netherlands
[3]Department of Geography, University of Zurich, Switzerland

**Correspondence:** Marit van Tiel (marit.van.tiel@hydrology.uni-freiburg.de)

**Abstract.** Warm and dry summer days can lead to low streamflow due to a lack of rainfall and increased evaporation. In glacier-ized catchments, however, such periods can lead to a very different hydrological response as glaciers can supply an increased amount of meltwater, thereby compensating for the rainfall deficits. Here, we analyzed glacier-fed streamflow responses to warm and dry periods (WD) in long-term streamflow observations (>50 years). WD events during summer (June – September) were analyzed for catchments with varying glacier cover in Canada, Norway and the European Alps. WD events were defined by days with temperatures above a daily varying threshold, based on the 80th percentile of the respective long-term temper-ature data for that day in the year, and daily precipitation sums below a fixed threshold (< 2 mm/d) for a minimum duration of seven days. Streamflow responses to these WD events were expressed as level of compensation ($C$) and were calculated as the event streamflow relative to the long-term streamflow regime. $C \geq 100\%$ indicates that increased melt could compensate, or even overcompensate, the rainfall deficit and increased evaporation. Results showed a wide range of compensation levels, both between catchments and between different WD events in a particular catchment. $C$ was, in general, higher than 100% for catchments with a higher relative glacier cover. June was the month with highest compensation levels, but this was likely more influenced by snowmelt than by glacier melt. For WD events in September, $C$ was still higher than 100% in many catchments, which indicated the importance of glacier melt as streamflow contributor in late summer. There was a considerable range in $C$ of different WD events for groups of catchments with similar glacier cover. This could be partly explained by antecedent conditions, such as the snow fallen in the previous winter and the streamflow conditions thirty days before the WD event. Some decreasing trends in $C$ were evident, especially for catchments in Canada and the European Alps. Overall, these results suggest that glaciers do not compensate straightforwardly. The different streamflow contributions and their variations are important for the buffering capacity and the compensating effect of glaciers in these high mountain water systems.

## 1 Introduction

Dry periods and heatwaves negatively affect water availability (e.g. Stahl et al., 2016; Teuling, 2018; van Loon, 2015; Zappa and Kan, 2007). Dry periods or meteorological droughts (i.e., relatively dry periods), alter the input of hydrological systems, while heatwaves or relatively warm periods increase evapotranspiration amounts if enough water is available (e.g. Mastrotheodoros et al., 2020; Teuling et al., 2013). In water balance terms, this means that during such combined dry and warm periods,





streamflow decreases and soil and groundwater storages are depleted. The summer of 2003 in Europe was an example of such a dry and hot period (Fink et al., 2004), which caused numerous negative effects; e.g. low water levels limiting transportation, reduced agricultural production, problems with water supply and forest fires (COGECA, 2003; Jonkeren et al., 2007; Rouault et al., 2006; Stahl et al., 2016). However, in some glacierized catchments in, for example, Switzerland and Austria, streamflow was above the long-term average during this extreme summer (Koboltschnig and Schöner, 2011; Koboltschnig et al., 2009;
Zappa and Kan, 2007). In high mountain regions, where snow and ice are present, these snow and ice storages provide an additional source of water, especially during warm periods, because of temperature-driven water supply.

Several studies have shown that the hydrological response of glacierized catchments in drought years stands out when analyzing a regional sample of catchments(e.g. Bakke et al., 2020; Zappa and Kan, 2007). While groundwater and snow are also known to be a buffer against meteorological droughts, these storages can also be themselves be depleted and considered to
be in a state of drought (e.g. Cooper et al., 2016; Hellwig and Stahl, 2018; Livneh and Badger, 2020; Van Loon and Van Lanen, 2012), which is sometimes referred to as 'groundwater drought' (Bloomfield and Marchant, 2013; Peters et al., 2005) and 'snow drought'(Huning and AghaKouchak, 2020). The buffer capacities of these three types of catchment storage (groundwater, snow and glaciers) differ. Groundwater has a delayed response to meteorological droughts and therefore, at the time of the event, can provide baseflow. Still, it does not provide extra water during warm and dry periods (compared to normal conditions). In
contrast, snow and glacier ice will provide more meltwater when temperatures are high. Therefore, snow and glaciers do not only act as a buffer, but their meltwater could also compensate the otherwise emerging streamflow deficit. For seasonal snow, however, there is a limited amount to melt, namely the snow that has accumulated in winter. Hence, groundwater and snow storages might not always be a perfect buffer during warm and dry periods. Glaciers are, theoretically, a favorable buffer during such periods because they generate extra melt when temperatures are higher than normal, and they do not get depleted on an
annual time scale.

The buffering effect of glaciers has been analyzed at different spatial and temporal scales. In general, studies indicate that glaciers provide an important source of water during warmer and drier periods in the year and during drought years specifically (e.g. Ayala et al., 2020; Jost et al., 2012; Kaser et al., 2010; Anderson and Radić, 2020). During such extreme drought years, runoff from glacier areas was estimated to contribute 55-100% of summer runoff in the Maipo river basin
in Chile (7.8% glacierized) (Ayala et al., 2020) and during the 2003 European drought and heatwave event, streamflow in glacierized catchments in the Alps was up to 40-60% higher than normal during August, depending on glacier cover fraction of the catchment and catchment elevation (Zappa and Kan, 2007; Koboltschnig and Schöner, 2011). For the whole High Mountain Asia region, Pritchard (2019) also found high relative monthly glacier melt contributions to streamflow in drought years, but mainly attributed this to a decrease in precipitation amounts. Often, these conclusions are drawn from modelling exercises that
allow separating the glacier melt contribution from other streamflow contributions such as snowmelt. However, modelling these glacierized hydrological systems is a challenge, because of the many intertwined hydrological processes (Finger et al., 2011; Konz and Seibert, 2010; van Tiel et al., 2020b). Hence, models may only give a rough estimation of the different streamflow contributions and may not adequately simulate the hydrological processes during such extreme warm and dry periods.





Other studies focused on the dampening effect of glaciers on the overall interannual streamflow variability (Fountain and
Tangborn, 1985; Rothlisberger and Lang, 1987). During warm and dry years, glaciers can provide more meltwater to stream-
flow, and during cold and wet years they generate less meltwater so that altogether the interannual streamflow variability is
relatively low. Pohl et al. (2017) found for the Pamir region that during years with strong negative anomalies of runoff and
snowmelt in the non-glacierized areas, glacier melt was high. And contrastingly, during a year with high precipitation amounts
and low temperatures, glacier melt showed negative anomalies. The result of the balancing between melt and precipitation is
assumed to depend on the catchment relative glacier cover (e.g. Chen and Ohmura, 1990; Fountain and Tangborn, 1985; Pohl
et al., 2017), but also other climate and catchment characteristics appear to influence the streamflow sensitivity to climatic
anomalies (Pohl et al., 2017; van Tiel et al., 2020a).

Glacier melt can thus be important to maintain streamflow during dry (and warm) periods. However, due to climate change
and warming, mountain glaciers have been retreating and will further do so in the future (e.g. Radić et al., 2014; Zemp et al.,
2015), affecting not only total downstream runoff but also compensation effects during dry periods. The negative glacier mass
balances that have been prevalent in recent years for all glaciers worldwide (e.g. Andreassen et al., 2005; Fischer et al., 2015;
Zemp et al., 2015) provided an additional source of water in the summer compared to the seasonally delayed contribution.
However, this source will not be sustained (e.g. Pritchard, 2019; Huss and Hock, 2018; Jansson et al., 2003). Several studies
have shown that some regions already show declining streamflow trends, while for other regions the moment of change from
increasing to decreasing trends (peak water) is projected in the future (Chesnokova et al., 2020; Huss and Hock, 2018; Moore
et al., 2020; Stahl and Moore, 2006). The question that remains open is whether we can also observe (already) a decrease in
glacier compensation capacity during warm and dry periods.

The glacier melt contribution to streamflow differs in different catchments, regions and times of the year. With the varying
glacier melt contribution to streamflow, also the compensation capacity varies between catchments. In general, highly glacier-
ized catchments have a higher relative glacier melt contribution to streamflow than low glacierized catchments. However, this
contribution may reach a maximum, as the higher glacierized catchments are generally located at higher elevations that receive
more orographic precipitation amounts (e.g. Koboltschnig and Schöner, 2011). The relative glacier melt contribution to stream-
flow is generally assumed to be highest in August and September, because earlier in the year snow is an important contributor
to streamflow too (Jost et al., 2012; Koboltschnig et al., 2009; Moore et al., 2020; Naz et al., 2014; Stahl et al., 2017). The
relative contribution is, however, also dependent on precipitation amounts and seasonality. For example, in Western Canada,
summer precipitation amounts are rather low, while in the European Alps they are relatively high, suggesting that glacier melt
is more important in Canada (Kaser et al., 2010; Viviroli et al., 2007). Also, catchment characteristics such as elevation and
storage capacity (groundwater), which relate to the hydrogeology, slope and size of the catchment, can influence the absolute
and relative contributions of glacier melt and other streamflow components. The effects of these differences among glacierized
catchments on the buffer effect and compensation capacity during warm and dry periods have not previously been addressed
systematically.

While most previous studies have looked at (a) relatively dry (drought) year(s) or summer(s) to analyze the glacier buffer
effect, we focus here on the hydrological response to specific warm and dry events, i.e. short periods that are characterized by





days with no to very low rain amounts and days with relatively high temperatures. This short time scale isolates the glacier compensation effect on streamflow and minimizes confounding other hydrological processes that might affect more aggregated signals. This scale enables investigating the compensation effect in different months during the summer and analyzing multiple periods to detect possible trends. Moreover, this scale is pivotal for water management because it will give insights in the streamflow response to extreme weeks (dry and warm) when downstream water availability can be low, water demand high and measures might be needed. In this study, we analyzed observed hydrological responses to WD events for catchments with varying glacier cover in Norway, Canada, Switzerland and Austria. The aims of the study were 1) to investigate how often and when such warm and dry events (WD) occur, 2) to analyze the general streamflow responses to these periods in different months, 3) to evaluate differences in levels of compensation between catchments and regions, and 4) to investigate differences in compensation levels between different WD events for individual catchments. Overall, we aim to give insights in which conditions glaciers compensate for the lack of rainfall-runoff and increased evaporation during warm and dry periods.

## 2 Data and hydroclimatology of selected glacierized catchments

### 2.1 Streamflow, meteorological and glaciological data

Daily streamflow (Q), precipitation (P) and temperature (T) data were obtained for 50 glacierized catchments in Norway (9), Canada (17) and the European Alps (24). These catchments were selected based on length of the time series (long records), a minimum amount of missing values and relative glacier cover ($gc$) (including low and high glacierized catchments). A few of these catchments are nested. Areal averaged precipitation and temperature data were derived from gridded data products: SeNorge2 for Norway (Lussana et al., 2016, 2018) (1 km x 1 km), PNWNAmet for Canada (Werner et al., 2019), (1/16° resolution, 7 km x 7 km), RhiresD and TabsD for Switzerland (MeteoSwiss, 2019, 2017) (2 km x 2 km), and SPARTACUS for Austria (Hiebl and Frei, 2016, 2018) (1 km x 1 km). Streamflow data were obtained from NVE (Norway), FOEN (Switzerland), eHYD (https://ehyd.gv.at/) (Austria), and the National Water Data Archive HYDAT (Canada).

The length of the available time series for the different catchments differed, between 50 and 68 years. P and T data were available for 1957-2015 for Norway, 1945-2012 for Canada, and 1961-2016 for the European Alps. For Norway and Switzerland, streamflow data were used until 2016, for Austria until 2015. For Canada, streamflow records ended between 2012-2017, but the selection of events was limited to 2012 because of the P and T data. Few data gaps present in the streamflow data occurred mostly in the winter months.

Glacier mass balance data for the different regions were obtained from the WGMS database (WGMS, 2019). Glacier mass balance data are measured on only a few glaciers worldwide, and even fewer glaciers have a long-term record (Zemp et al., 2009). Therefore, regional average mass balance time series were calculated from all available mass balance observations per year per country (Austria, Switzerland, Norway and Canada) by taking the median. Some glaciers have only annual mass balance measurements, while others also have seasonal (summer and winter mass balances). Since these mass balance measurements represent the integrated response over a year or season, they are not directly applicable to analyze the compensating





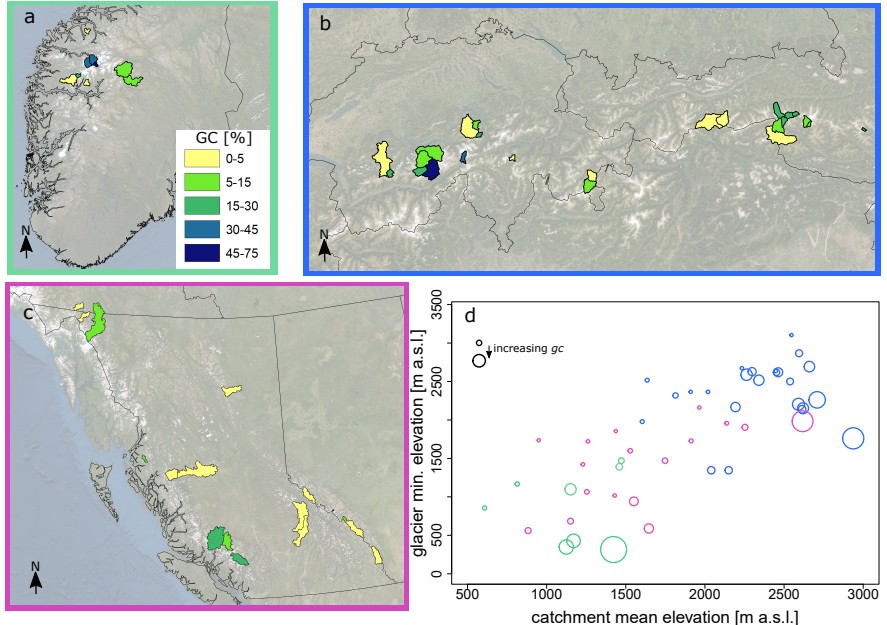

**Figure 1.** Location of the glacierized catchments in a) Norway, b) European Alps and c) Canada and catchment elevation characteristics (d).

effect of glaciers on the short time scale. Still, they might give additional directional information about the glacier response to WD events in different years.

Glacier cover data for the catchments were obtained from the Austrian glacier inventory (GI4, 2015), the Swiss Glacier inventory (2010) (Fischer et al., 2014), the NVE Landsat outlines for Norway (1999-2006) (Andreassen et al., 2012), and the
130 Randolph glacier inventory (RGI) for Canada, representing glacier cover around 2005 (Pfeffer et al., 2014). These shapefiles were used to calculate the percentage of each catchment that is glacier-covered.

## 2.2 Catchment characteristics

The Norwegian study catchments are located in the southern part of Norway, close to the Atlantic Ocean (Figure 1a). The catchment with the highest $gc$ of all the sample catchments is situated in Norway; the Nigardsbrevatn catchment, with a $gc$ of
135 71.7%. In general, the Norwegian catchments have the lowest mean elevations (Figure 1d) and are located at higher latitudes than the Canadian catchments. Catchment sizes range from 65 km$^2$ to almost 800 km$^2$.

The Canadian catchments are located along the west coast and in the western interior part of Canada. Most of the catchments are situated in the province of British-Columbia, a few in Alberta and one in the Yukon Territories (Figure 1c). The catchment with the highest $gc$ is the Sunwapta River in the Rocky Mountains, with 55.5% glacier cover. This catchment is hardly visible
140 on the map in Figure 1, because it is also the smallest catchment (29.3 km$^2$). Overall, the catchments in Canada are much larger than the catchments in the Alps and Norway, with the other catchments ranging in size from 250 km$^2$ to 6860 km$^2$. The Canadian catchments span a broad range of mean catchment elevations and glacier elevations (Figure 1d).





The catchments in Switzerland and Austria are distributed over the Alps from west to east (Figure 1b). The Massa catchment in Switzerland has the highest $gc$, 56.5%. Catchments in the European Alps are generally situated at higher elevations, and also the minimum glacier elevation is highest compared to the other regions (Figure 1d). Catchment areas range from 9 km$^2$ to 380 km$^2$.

## 2.3 Hydroclimatology

The selected glacierized catchments are located in different regions, different mountain ranges, and different latitudes, elevations and proximities to the ocean. They also have different $gc$ and catchment sizes. Altogether this results in a large variation of precipitation seasonalities and amounts (Figure 2) and streamflow regimes (SI Figure 1) among the studied glacierized catchments, possibly influencing the catchment responses to WD events.

Total precipitation, as well as its distribution over the year, vary within and across the three regions (Figure 2). In Norway, most catchments show high precipitation amounts in winter and lower amounts in spring and summer (Figure 2a). The SeNorge2 precipitation product for Norway is known to underestimate precipitation in mountainous regions due to sparse observations (Lussana et al., 2018), so total monthly amounts are likely higher than plotted here. However, the distribution over the year and the differences between the Norwegian catchments are assumed not to be influenced by that. Norway has a strong west-east gradient in climate continentality (e.g. Engelhardt et al., 2014). The two catchments located most inland receive much less precipitation and have less monthly variation in precipitation amounts over the year (Figure 2b).

The Canadian catchments show the largest variation in precipitation regimes. Four precipitation types were classified: 1) a mixture of precipitation regimes that show low variations in precipitation amounts over the year, but mostly a small peak around June (Figure 2c), 2) a regime where precipitation is lower in summer and high in winter (Figure 2d), 3) a regime with lowest precipitation in April and highest precipitation in winter, but less intra-annual variation as in regime 2 (Figure 2e), 4) high precipitation in summer (Figure 2f). Catchments with type 1 precipitation are located in Alberta, on the eastern side of the Rocky Mountains. Type 2 catchments are located in the southern part of British-Columbia and along the coast. The catchment that receives much more winter precipitation than the other catchments is the small Exchamsiks River catchment in the Coast Mountains. The three most northern catchments, together with the Nautley River catchment (which has the lowest elevation of the Canadian catchments), have precipitation type 3. The one catchment that is located in the northern Rocky Mountains has precipitation type 4.

In the European Alps, most catchments have somewhat higher precipitation amounts from May until August. In September, the rainfall amounts are relatively low (Figure 2g). Two catchments with a more even distribution of precipitation amounts over the year are located in central Switzerland (Figure 2i). Two catchments that show higher precipitation amounts from May until November, but generally have lower amounts compared to the other catchments are located in the drier south-eastern part of Switzerland (Figure 2h).

Temperature regimes were more similar across the catchments (not shown), with temperatures well below zero in winter and above zero in summer. On average, the Norwegian catchments have the shortest, whereas the Canadian catchments have the most extended season with above-zero temperatures. The Canadian catchments in general also have a more considerable



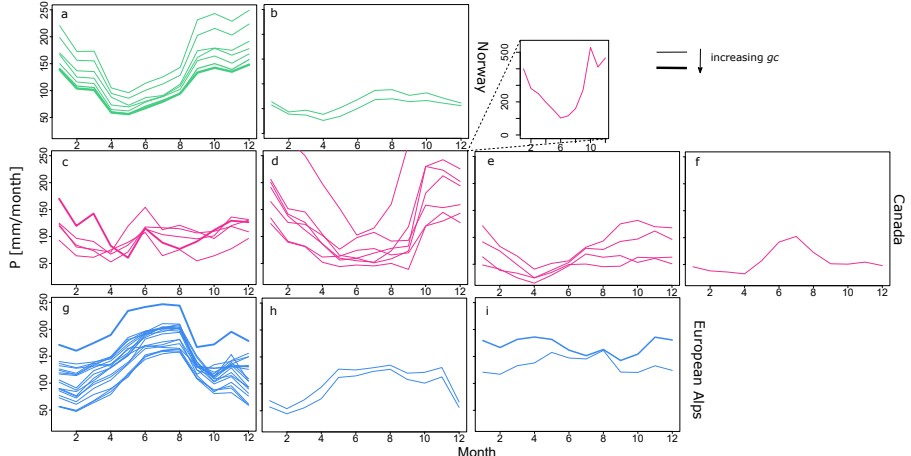

**Figure 2.** Precipitation seasonality of the glacierized catchments in Norway (a & b), Canada (c, d, e, & f) and the European Alps (g, h, i). The graph connected to d shows a different y-axis scale for one catchment that had large monthly precipitation amounts.

difference between winter and summer temperatures. The time of year when monthly mean temperatures reach above-zero values is later for most of the catchments in the European Alps and in Norway than for the Canadian catchments.

## 3 Methods

### 3.1 Selection of events

WD spells were selected based on a daily precipitation and temperature threshold. Days were defined as dry when the 7-day moving average precipitation sum did not exceed 2 mm/d. We used an absolute definition of dry, and not a relative definition (as is common when studying meteorological droughts), to focus on processes in the catchment that happen when there is no (or very little) rain input and thus hardly any rainfall-runoff generation. We did not use a precipitation threshold of zero, because due to the gridded interpolated datasets, even if only a tiny part of the catchment receives a little amount of rain, the catchment average precipitation does not equal zero.

The temperature threshold was based on a 7-day moving positive degree-day sum (DD7), from which the 80th percentile was calculated with a 30-day moving window (similar to seasonally moving drought thresholds, e.g. van Loon, 2015)). In contrast to the fixed precipitation threshold, the temperature threshold is a relative threshold, varying for each catchment and for each day of the year to capture anomalies from the normal seasonal cycle. This means that a defined warm event in June is not necessarily comparable in absolute temperatures or degree day sums with a warm event in July. The temperature thresholds were based on temperature data for the period 1961-2010 for all regions.

Days were selected when both the 7-day moving average daily precipitation was below the precipitation threshold, and the DD7 was above the temperature threshold. When these days were consecutive, or when there were one or two days between





**Table 1.** Definitions of the different dry events

| Dry event | Acronym | P threshold | T threshold |
|---|---|---|---|
| Cold & Dry | CD | <2 mm/d | DD7 < $20^{th}$ percentile |
| Normal & Dry | ND | <2 mm/d | $40^{th}$ percentile < DD7 < $60^{th}$ percentile |
| Warm & Dry | WD | <2 mm/d | DD7 > $80^{th}$ percentile |
| Extreme Warm & Dry | WWD | <2 mm/d | DD7 > $90^{th}$ percentile |

not meeting the conditions but the (daily) precipitation during these days was smaller than 5 mm/d, these days were counted as one event. Events were selected that had a minimum duration of 7 days and had the mid-date of the event from June to September (see SI Figure 2 for an example).

To investigate the role of temperature during dry periods in general, not only warm and dry (WD) events were selected, but also dry and relatively cold (CD), dry and normal temperature (ND), and dry and extreme warm events (WWD), by varying the temperature threshold percentile (Table 1). WWD events are part of WD events, but are in general shorter in duration.

### 3.2 Streamflow response and level of compensation

Once the periods were selected based on the P and T data, the corresponding streamflow data were analyzed. It was assumed that in these high elevation catchments there is an immediate response to the WD events (or the other dry events) and streamflow data of the dates exactly corresponding to the events were selected. Since the events last at least 7 days, we assume that within these 7 days there is a clear response to the WD event. For each of the selected events a compensation metric $C$ was derived (equation 1).

$$C = \frac{\sum_i^j Q}{\sum_i^j Q_n} \cdot 100\% \tag{1}$$

in which $C$ is the level of compensation (in %), Q [mm/d] is the 7-day smoothed daily streamflow during the event, Qn [mm/d] the long-term daily streamflow regime as a benchmark and i and j the start and end dates (or DOY for Qn) of the event, respectively. Qn is based on the whole time series, but starting with the start of the respective P and T data (1961 for the European Alps, 1957 for Norway, and 1945 for Canada – if the streamflow time series start later, then that period was selected to calculate the regime). The long-term daily averages were then smoothed with a 7-day moving average, like the P, T and Q data, to obtain Qn. $C$ values above 100% indicate overcompensation of the rainfall deficit by excess melt because streamflow was higher than the long-term average. Values below 100% indicate a low level of compensation, i.e. streamflow during the event was below normal conditions and the melt of snow and ice could not completely compensate for the lack of rainfall and increased evaporation. $C$ of the events were grouped according to month and region.

The streamflow trend during the event was also analyzed, i.e. whether streamflow was increasing or decreasing during the WD period. The latter was done by a linear regression model relating streamflow to time. The slope of the regression model





was used as proxy for the general trend of streamflow during the event. When there is no precipitation input for the catchment, streamflow usually recedes, but melt input from snowpack or glacier could cause an increase in streamflow.

### 3.3 Controls and drivers of variability in the level of compensation

To investigate which factors control the variability in $C$ 1) on the catchment scale, and 2) on the event-to-event scale, several characteristics of each catchment and of each WD event were extracted (Table 2).

**Table 2.** Catchment and event characteristics that were used to explain the variability in the level of compensation. ($C$)

| | Variable | Description | Hypothesized relation with $C$ |
|---|---|---|---|
| Catchment characteristics | $gc$ | Catchment relative glacier cover | Higher $gc$: larger relative glacier melt contribution and smaller relative rainfall contribution –> less sensitive to dry periods (higher $C$). Also larger temperature-sensitive streamflow contribution. |
| | $E_c$ / $E_g$ | Mean elevation of the catchment/ glacier(s) | Higher elevation: more precipitation (lower $C$) and lower temperatures (lower $C$) and relatively more snow storage (lower/higher $C$). Also, evaporation losses might be lower (higher $C$) and available soil storages smaller (lower/higher $C$). Higher elevated glaciers: lower temperatures (lower $C$) and later start of ice melt (lower $C$). |
| | $E_r$ | Elevation range catchment | Large elevation range: large temperature gradient and several hydrological processes occurring at the same time, such as melt (higher $C$), evaporation (low $C$) and snow storage (higher $C$/lower $C$). |
| | $E_{g-}$ | Minimum elevation of the glacier(s) | Lower elevation glacier tongue: more and earlier in the season glacier melt (higher $C$). |
| | $P_{summer}$ | Mean precipitation sum of July and August | Low rainfall amounts: higher importance of glacier melt (higher $C$). Catchments in the European Alps that were relatively wet in summer (>300 mm) and Canadian catchments that were relatively dry (<200 mm) were compared. |
| WD event characteristics | $T_{spring}$ | Percentile of mean temperature in spring (MAM) | Proxy for early disappearance of snow at lower elevations. Lower $C$ for WD event in early summer. Higher $C$ if less snow on the glacier reduces albedo and increases glacier melt |
| | $P_{winter}$ | Percentile of sum of precipitation in winter (DJF) | Proxy for snow amount. Higher $P_{winter}$ –> more snowmelt early summer (higher $C$) and more snowmelt recharge that is released later in the year (higher $C$). Contrary, more snow covers the glacier for longer and results in less glacier melt (lower $C$). |





| $Q_{30}$ | The streamflow percentile of the 30 days before the WD event | Proxy for the state of the catchment storages before the event. High $Q_{30}$ –> filled storages (higher $C$). Additionally, it may indicate that the event takes place in a season or year that is in general characterized by higher than normal flows (e.g. extreme heat summer). |
| $T$ | T anomaly during event. DD7/DD7$_{average}$ | Higher T –> higher $C$ |
| $D$ | Duration | Longer events: Precipitation deficits accumulate, and storages deplete (lower $C$). The glacier drainage system may become more efficient (higher $C$). |
| $MB$ | Regional mass balances of the year the WD event takes place | Higher $MB_{sum}$ (more negative MB) –> higher glacier melt contribution (higher $C$). Higher $MB_{win}$ –> more snow (lower/higher $C$). |
| $Y$ | Year the WD event takes place | Test the presence of a trend. Have compensation effects been reduced due to retreating glaciers (lower $C$)? |

The catchment characteristics were used in a linear model to analyze which control explains most of the variability in $C$
across the different catchments for each month and region. The relative importance of the catchment controls was assessed by
the proportion of variance explained by each predictor by averaging the added explained variance over all possible orderings
(calculated with the R-package Relaimpo (Grömping, 2006)). The same method was applied to analyze the importance of
different event drivers. For this analysis, catchments were grouped by glacier cover, and for each month, each region and each
glacier cover group a regression model was set up. A minimum of 10 events in each group was required.
To assess trends in $C$ and examine the relation between $C$ and the regional glacier mass balance observations, Spearman rank
correlation coefficients were calculated. A coefficient was calculated if there were at least 8 or more events for the respective
catchment in the respective month.

## 4 Results

### 4.1 Occurences of events

WD events occurred in all summer months (June until September) and in all regions (Table 3. Catchments in the European
Alps had most of the events in July and September, while in Norway and Canada there were more events in June and July.
The number of events differed per catchment, with some catchments having more than double or three times the number of
events compared to the catchment with the least number of events. In Canada, one catchment, in particular, had very few events
(located in the northern part of the Rocky Mountains), hence the low numbers in the table for Canada. The numbers are not





**Table 3.** Number and duration of WD events. The numbers in brackets indicate the range of numbers/durations for the individual catchments

| Region | # June | # July | # August | # September | Years most events | Duration of events [d] |
|---|---|---|---|---|---|---|
| Alps | 86 (1-9) | 130 (2-10) | 119 (1-9) | 234 (5-14) | 1982, 1983 | 10.2 (7-21) |
| Norway | 166 (16-22) | 133 (11-16) | 109 (10-13) | 85 (4-14) | 2002, 2006 | 10.8 (7-28) |
| Canada | 241 (7-19) | 266 (2-23) | 190 (1-19) | 151 (0-19) | 1967, 1990 | 10.6 (7-40) |

**Table 4.** Number of CD, ND and WWD events. The numbers in between brackets indicate the range of number of events for the individual catchments.

| Region | Type | # June | # July | # August | # September |
|---|---|---|---|---|---|
| Alps | CD | 13 (0-2) | 3 (0-1) | 1 (0-1) | 66 (0-7) |
| | ND | 22 (0-2) | 7 (0-2) | 20 (0-4) | 35 (0-3) |
| | WWD | 43 (0-6) | 80 (0-6) | 53 (0-5) | 111 (0-9) |
| Norway | CD | 35 (1-7) | 15 (0-5) | 16 (0-3) | 16 (0-4) |
| | ND | 18 (0-4) | 12 (0-4) | 7 (0-1) | 13 (0-5) |
| | WWD | 91 (6-11) | 64 (4-7) | 58 (4-7) | 26 (1-4) |
| Canada | CD | 52 (0-9) | 24 (0-6) | 41 (0-8) | 32 (0-5) |
| | ND | 50 (0-8) | 68 (0-8) | 108 (0-10) | 68 (0-6) |
| | WWD | 136 (2-11) | 132 (0-10) | 97 (0-7) | 37 (0-5) |

comparable between the regions and catchments, because of a different length of the time series and different numbers of catchments per region.

Most of the events in the European Alps occurred in the year 1982 (34 events in total in all catchments) and 1983 (38 events). For the Norwegian catchments, most events occurred in 2002 (28 events) and in 2006 (34 events). In Canada, the years with many events are earlier in the observed time series, namely 1967 (36 events) and 1990 (37 events).

The mean duration of the events was highest in Norway, then Canada and then the Alps. The longest event occurred in a Canadian catchment and lasted 40 days. Most of the events had a relatively short duration, between 7 and 10 days. For the Alps, the mean duration was longest in September (11.4 days) and shortest in June (9.0 days). In Norway, the events were on average longest in June (12.1 days) and shortest in July (9.0 days). In Canada there was less difference between the mean durations in the different months, in September the average duration was 10.1 days, and in June and July, it was 10.7 days on average.

Analyzing the effect of temperature by selecting other dry events, shows that WD conditions occurred more often than ND, or CD conditions (Table 4). The WWD events partly include the WD events and occurred more often than the ND and CD events. In the Alps all different dry events mostly occurred in September. In Norway, most events occurred in June. In Canada, WWD events occurred mainly in June and July, ND events in August, and CD events in June.



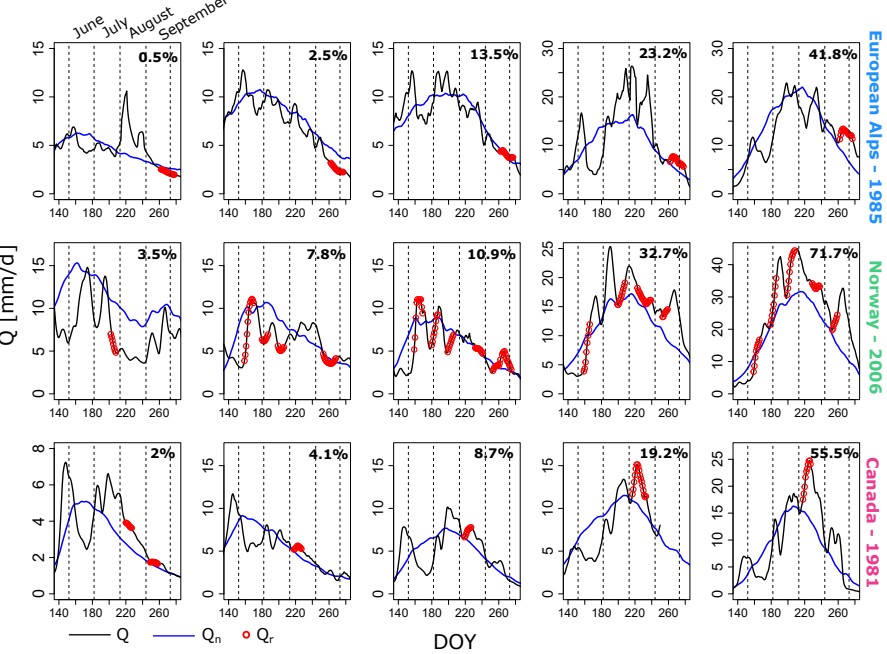

**Figure 3.** Examples of streamflow responses (Qr) to WD events in different catchments and in different months. Each row represents one region and one specific year. The columns are different catchments, they are sorted from low to high relative glacier cover (left to right). Note the different y-axis for some of the plots.

## 4.2 Glacier compensation in different catchments

In general, $C$ showed a wide range of values for the different WD events. Streamflow can show an increasing trend during the event, or decreasing, or both. Examples of streamflow behavior during WD events for different catchments with varying relative glacier covers show the importance of glacier cover (Figure 3). The September event in the European Alps in Figure 3 showed that streamflow was above normal for the two catchments with >23.2% $gc$ and close to normal for the catchment with 13.5% $gc$. Streamflow was more below normal ($C < 100\%$) for the catchment with 2.5% $gc$ compared to the catchment with

only 0.5% glacier cover, possibly indicating that other factors than $gc$ play a role too. For Norway, the events in July showed different trends, with increasing $Q$ during the event for the three catchments with the highest $gc$ in the example and decreasing $Q$ for the two lower $gc$ catchments (Figure 3). Only for the two highest glacierized catchments in this example, $C$ was larger than 100%. In Canada, the example in Figure 3 shows for an event in August that streamflow is mostly pushed above $Q_n$ in all catchments with different $gc$. Overall the examples also show that sometimes events are embedded in a longer positive or

negative anomaly of streamflow compared to the regime (Canada catchment 2% $gc$ and Norway catchment 3.5% $gc$).

Regressing $C$ for a particular month and region against the catchment characteristics as independent variables showed that $gc$ was the most important variable to explain $C$ in all summer months in the European Alps, and in August and September for the Canadian catchments, and in August for the Norwegian catchments (Table 5). Mean catchment elevation was often the





**Table 5.** Variance in $C$ explained by catchment characteristics in a linear regression model. n indicates the number of WD events that were used for the regression.

| Region | Month | n | total var. explained [%] | $gc$ [%] | $E_c$ [%] | $E_g$ [%] | $E_{g-}$ [%] | $E_r$ [%] |
|--------|-------|----|-----|------|------|------|------|------|
| Alps | Jun | 86 | 41.1 | 23.0 | 10.4 | 3.6 | 3.1 | 1.0 |
| | Jul | 130 | 31.5 | 13.8 | 10.3 | 4.9 | 1.7 | 0.8 |
| | Aug | 119 | 60.8 | 28.2 | 19.4 | 9.4 | 2.3 | 1.5 |
| | Sep | 234 | 40.2 | 21.0 | 8.6 | 7.5 | 1.8 | 1.4 |
| Norway | Jun | 166 | 14.4 | 5.5 | 4.0 | 2.0 | 2.5 | 0.5 |
| | Jul | 133 | 29.2 | 8.9 | 4.5 | 3.2 | 7.0 | 5.6 |
| | Aug | 109 | 33.1 | 10.1 | 7.0 | 5.6 | 5.9 | 4.7 |
| | Sep | 85 | 57.2 | 9.5 | 17.1 | 17.1 | 4.1 | 9.4 |
| Canada | Jun | 241 | 13.1 | 2.3 | 5.6 | 1.9 | 1.8 | 1.5 |
| | Jul | 266 | 13.0 | 5.7 | 2.9 | 1.2 | 1.8 | 1.5 |
| | Aug | 190 | 33.0 | 13.3 | 9.9 | 4.8 | 2.4 | 2.5 |
| | Sep | 151 | 33.7 | 27.2 | 2.7 | 1.1 | 2.0 | 0.7 |

second most important variable. For September in Norway, $C$ was best explained by a combination of mean elevation of the
catchment and the glacier. However, these catchment characteristics could often only explain part of the variance of $C$ in each
month and region (i.e. often less than 40% explained). All variables, but minimum glacier elevation ($E_g-$) showed a positive
relation with $C$.

The higher $gc$, the higher the streamflow was above Qn during an event (Figure 4). The spread in $C$ was, however, large.
Catchments in the European Alps covered almost the complete range of $gc$ and showed a strong relation between $gc$ and mean
catchment $C$ for all months. On average, the streamflow there was below the normal regime for catchments with $gc$ <10%.
In Norway, some of the catchments showed a relatively low $C$ in September. The higher glacierized catchments showed a
range of $C$ that varies from below 100% to above 100%, while for higher glacierized catchments in the European Alps and
Canada mostly all of the compensation levels were above 100% (overcompensating). In Canada, even the very low glacierized
catchments showed on average $C$ that were above 100%. The spread in responses was smallest in August in all regions and
larger in June and September. $C$ was on average highest in June for all regions.

The direction of the trends showed substantial streamflow increases during the events in June and July, especially for the
higher glacierized catchments (Figure 5). In August and September, the increased melt during the event compensated for the
general tendency of decreasing streamflow in these times of the year, resulting in a slope around zero instead of negative slopes.
For the few highly glacierized catchments in all the three regions, the mean trends were even slightly increasing in August and
September.





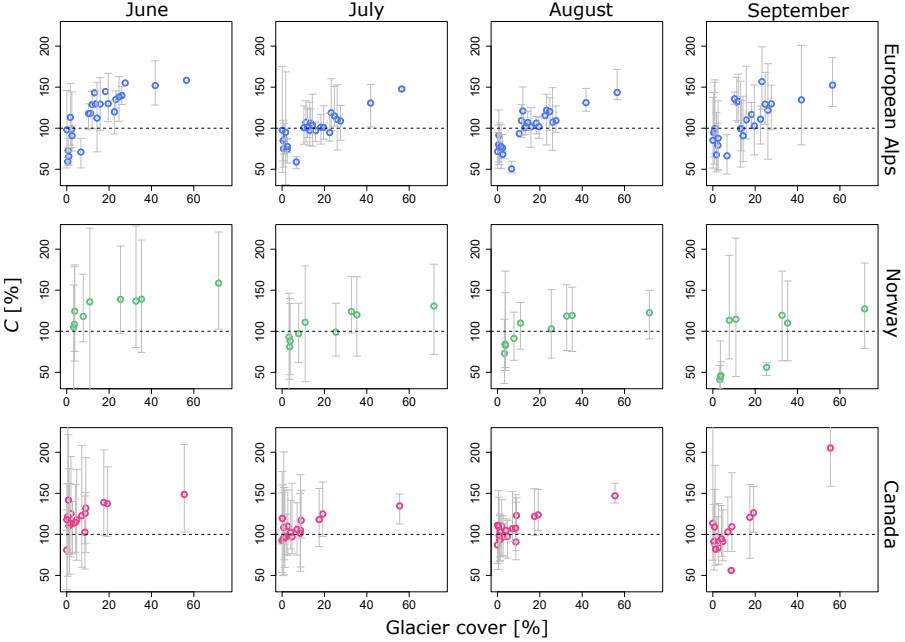

**Figure 4.** Level of compensation ($C$) during WD events for the different regions (rows) and the different months (columns) against $gc$ of the different catchments. The coloured circles indicate the mean response and the gray bars show the range of $C$, i.e. the minimum and maximum $C$.

### 4.3 Glacier compensation during other dry periods

The streamflow responses of WWD and WD events were opposite to those of CD events (Figure 6), although the number of CD events is low and therefore cannot be compared with WD responses in all months. During CD events, $C$ was below 100% for most of the catchments, except for some low glacierized Canadian catchments. The WWD events resulted in higher

$C$ compared to $C$ during WD events from July to September, but not in June. For the lower glacierized catchments (< 10%) the WWD event compensation levels were below the normal regime, mostly in Norway and the European Alps. The ND $C$ values were below 100% or close to 100%, indicating the effect of a rainfall deficit without the aid of excess melt due to high temperatures. Due to the limited number of events, or even absence of certain event types, the month in which the $C$ of the various dry events differs most clearly, i.e. when excess melt can make the most difference, could not be determined.

Different rainfall amounts could be another control of the catchment response to dry periods. Catchments in the Alps have relatively high rain amounts in summer, while some of the Canadian catchments have low rain amounts in summer. Comparing these two sets of catchments showed that the Canadian catchments have higher $C$ compared to catchments in the Alps, during WD events (Figure 7). Having less rainfall in general might indicate that during dry events, the relative rainfall deficit is smaller, and the relative glacier melt contribution is larger. However, other variables vary as well, e.g. the Canadian catchments have a

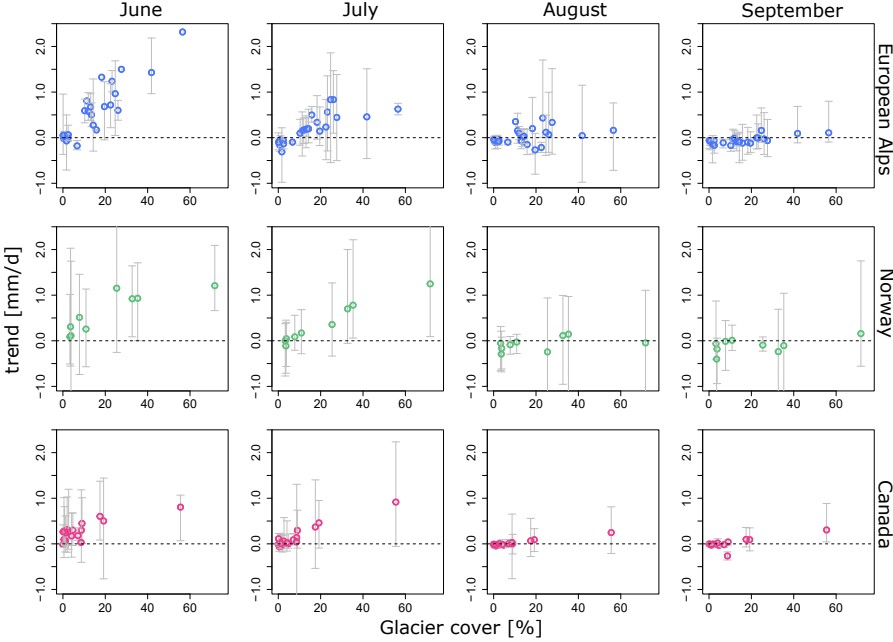

**Figure 5.** Trend of Q during the WD in different regions (rows) and months (columns). The coloured circles indicate the mean trend during the events per catchment and the grey bars represent the whole range.

lower glacier and catchment mean elevation compared to the ones in the Alps. Also, the relative glacier covers of the Canadian and Alps catchments are more complementary than comparable in this sample of catchments.

### 4.4    Drivers of event-to-event variability in compensation levels

Besides differences among catchments, there were also differences in $C$ among the selected WD events in individual catchments. The gray bars in Figure 4 show these sometimes rather large ranges. To explain these ranges, the relation between $C$
and several other variables was therefore tested for groups of catchments with similar glacier cover.

     Together, these variables could explain up to 80% of the variance of $C$, but more often they explained around 40-60% of the variance (Figure 8). Taking all the $C$ for the different summer months together considerably lowered the explained variance, especially for the Norwegian catchments, suggesting that the variables can have a different effect or represent different processes in the different months. The two variables that were used as a proxy for snowpack and snowmelt, $P_{winter}$ and $T_{spring}$
appeared to be most important in Norway, especially in June and July. In August and September, the most important variable switched to temperature in the Norwegian catchments. In the Alps, $P_{winter}$, $T_{spring}$ and antecedent streamflow conditions ($Q_{30}$) were important. In August and September, the snow variables were still important for the higher glacierized catchments. In Canada, $Q_{30}$ was the most important variable in all months and for most glacier cover classes. Most of the variables had



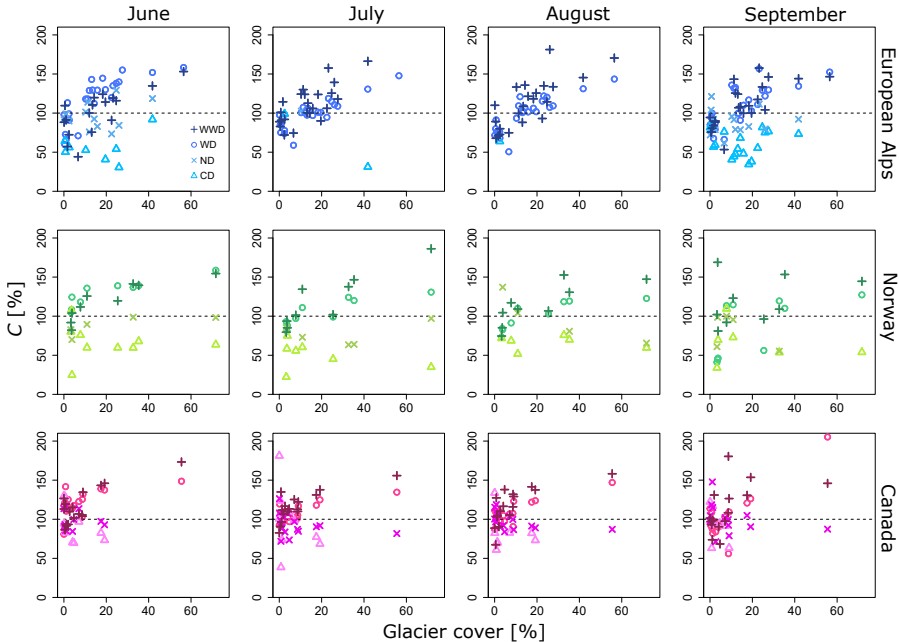

**Figure 6.** Mean catchment level of compensation ($C$) during WWD, WD, ND and CD events. If all dry periods occurred, one catchment shows four symbols in a vertical line.

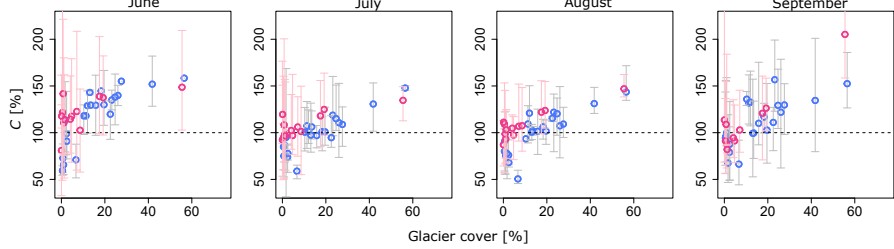

**Figure 7.** Level of compensation ($C$) during WD events compared among catchments that have low summer rain (July and August) (Canada, pink) and high summer rain amounts (European Alps, blue).

a positive relation with $C$, the higher the anomaly in the predictor, the higher $C$, except for $T_{spring}$ and sometimes duration, which had a negative relation. Higher temperatures in $T_{spring}$ would result in lower $C$.

Observed glacier mass balances provide information on the glacier melt contribution and snow accumulation on the glacier in different years. For most catchments, the correlations of $C$ with regional mass balance observations are quite low, and the range in correlations for catchments with similar glacier cover can be large (Figure 9). Annual and winter regional glacier mass balances showed mostly positive correlations with $C$, especially for the lower glacierized catchments. Winter mass balances ($MB_{win}$) were positively correlated with $C$ in June and July in Norway. Even at the end of summer, in September,





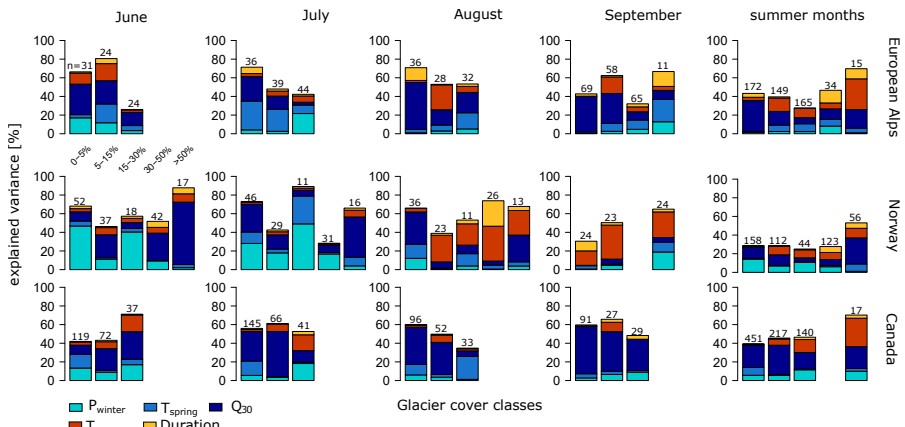

**Figure 8.** Explained variance of $C$ during WD events for groups of catchments with similar glacier cover. No bar is present if there were less than 10 events for a $gc$ class and month. The number above the bars indicate the number of events used in the regression model.

some catchments in Canada and European Alps showed a positive correlation between $C$ and $MB_{win}$. In July, August and September, the Norwegian catchments showed a switch from positive to negative correlations between $C$ and $MB_{sum}$ when moving to catchments with a higher $gc$. A negative correlation means here that $C$ is higher when $MB_{sum}$ is larger (more negative). Contrastingly, in Canada and the European Alps, in September, the correlation switched from negative to positive
with increasing $gc$.

Changes in glacier compensation effects over time due to glacier retreat might explain differences in $C$ between different WD events. For all the catchments with enough events to calculate a trend, only very few showed significant trends (Figure 10). Most significant trends were found in June (Canada) and September (European Alps), which were all negative. Norwegian catchments showed mostly positive trends, except in September.

## 5 Discussion

### 5.1 Quantifying the buffering capacity and compensation effect of glaciers

The presence of glaciers in headwater catchments indicates that potentially streamflow of these catchments can have an opposite response to warm and dry events compared to non-glacierized catchments. In general, glacier (and snow) melt can be seen as an additional source of water in summer, besides rainfall. Thus, during such warm and dry events, glaciers always alleviate
the (negative) hydrological response and they buffer against the negative impact of these meteorological conditions. In this perspective, the benchmark to compare the streamflow response is a situation without glaciers. Often, studies use this perspective to describe the buffering capacity of glaciers (e.g. Frenierre and Mark, 2014; Pritchard, 2019). However, quantifying or differentiating the buffering capacity, in this case, is not possible as it is either buffering (glaciers present) or not (no glaciers (anymore)). Therefore, we investigated in more detail the level of streamflow compensation to quantify the (active) buffering

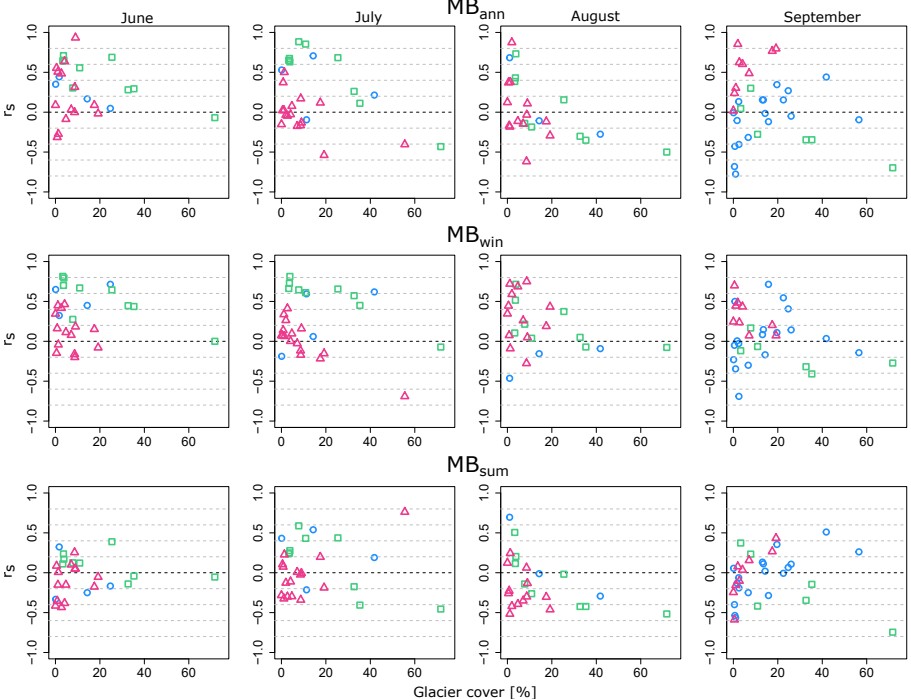

**Figure 9.** 9 Spearman rank correlation between $C$ and $MB_{ann}$ (upper row), $MB_{win}$ (middle row) and $MB_{sum}$ (lower row) for events in different months (columns). Colours and symbols indicate the three regions (as in other figures).

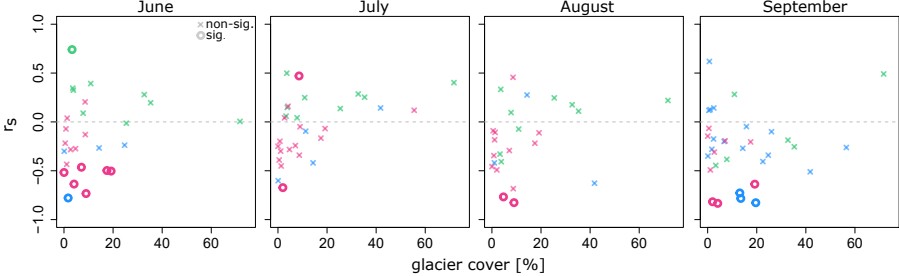

**Figure 10.** Time trends of $C$ calculated as Spearman Rank Correlation Coefficients ($r_s$). Circles indicate significant trends (a=0.05).

role of glaciers. For this, we used a benchmark that includes the long-term average glacier melt contribution (daily regime benchmark). We asked if the excess (more than normal) glacier melt during such an event can compensate for the reduced streamflow because of a lack of rainfall and potentially increased evaporation. The level of compensation ($C$) metric that we used can provide information on the water availability situation and distinguishes the buffering role of glaciers in different catchments and during different events. Such a metric that quantifies the buffering role and compensation effects of glaciers

is highly needed, to compare different studies, different situations and to analyze changes over time. The latter might present





some additional challenge as in these rapidly changing systems. The daily regime benchmark can change significantly over long periods (e.g. Van Tiel et al., 2018).

We thus looked at relative streamflow amounts instead of absolute amounts of streamflow and glacier melt contributions. This relative anomaly-based approach regarding temperature and streamflow allowed us to select events in different months. An absolute temperature threshold would only result in events in high summer. From a catchment and water management perspective, it is essential to understand how glaciers can buffer such events in the summer shoulder seasons, thus when meteorological conditions are exceptional compared to the normal condition.

## 5.2 Glacier cover as important control of compensation

Glacier cover was hypothesized to be an important control of the event compensation levels as it relates to the fraction of streamflow from glacier melt and thus determines the part of streamflow that is sensitive to temperature anomalies (Figure 11). In general, the results confirmed this hypothesis, especially for catchments in the European Alps. These findings correspond with the findings of Zappa and Kan (2007). They found for the extreme summer of 2003 close to normal conditions for glacierized catchments in the Swiss Alps with 10-25% glacier cover and close to 160% streamflow for a catchment with more than 60% glacier cover. Our event responses for the same highest glacierized catchment are of the same magnitude (150%), suggesting that compensation levels for events and longer-term drought responses may be similar. Koboltschnig and Schöner (2011) also found that above a glacier cover of around 10%, August 2003 streamflow was higher than average August streamflow, for glacierized catchments in the Drau, Salzach and Inn basins in Austria. Bakke et al. (2020) looked at another drought event (2018) and suggested that in Norway, catchments with glacier cover above 30% showed above normal streamflow conditions during August and September, but they did not specifically quantify how much. Overall, relative glacier cover is thus a first-order variable that determines the relative streamflow response. This means that in downstream areas, where the relative glacier cover reduces (less than 5%), most of such events may not be compensated by excess glacier melt. But Figure 4 shows that there are exceptions.

## 5.3 Drivers of event-to-event variability in compensation levels

A most notable finding was the large event-to-event variability in compensation levels. While some variation was expected, because of varying event characteristics (duration, temperature, timing within the month, small precipitation amounts) compensation level differences larger than 50%, often spanning below and above normal conditions were not necessarily expected. These wide ranges also make it difficult to answer when glaciers compensate because it depends on the situation. Understanding these variations, why sometimes the glacier is compensating and sometimes not, is crucial for water management purposes. The range in event responses was in general largest for the Norwegian catchments. In August, the range was relatively small in all regions, possibly indicating that in this month, other streamflow contributions than glacier melt are less important and thus cause less variability (Figure 11). August is also in other studies often described as the month with the highest relative glacier melt contributions (e.g. Stahl and Moore, 2006; Moore et al., 2020). June and September may show high event-to-event variability because of varying snow conditions in June and variations in end-of-season conditions in September (prolongation





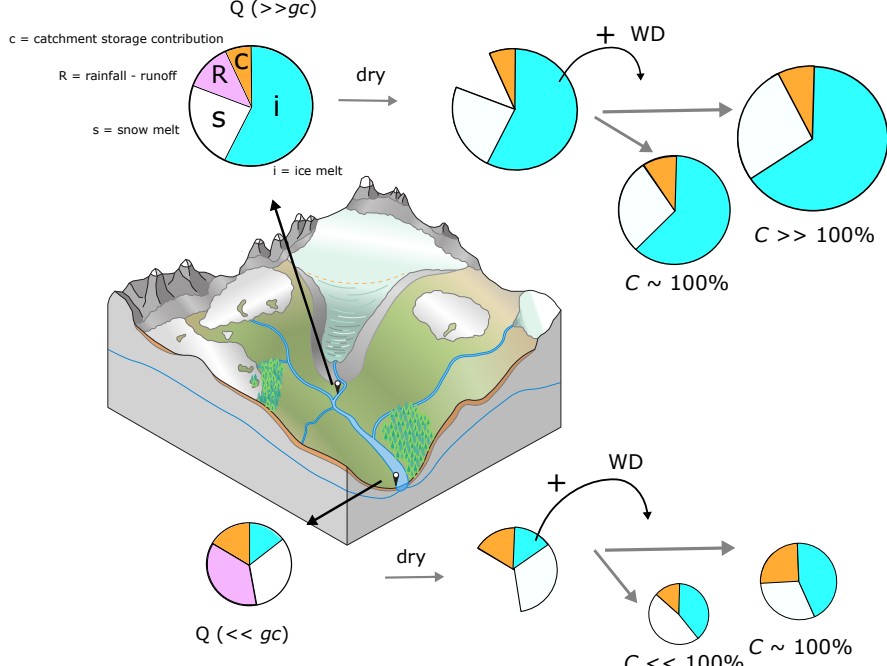

**Figure 11.** Conceptual overview of the different streamflow contributions in a glacierized catchment and how they influence the response to warm and dry events (WD). The sketch in the middle shows a glacierized catchment and the circles above and below represent the streamflow contributions for a highly glacierized (above) and a low glacierized catchment (below). Size of the circles scale with absolute streamflow amounts. The colors in the circles indicate the relative contributions of ice melt, snow melt, rain and other catchment storages. During a dry event, the rainfall-runoff contribution is lacking and depending on the other streamflow components and their anomalies glacier melt can compensate ($C$=100%) or overcompensate ($C$>100%) the deficit.

of August melt conditions or early onset of receding to baseflow conditions). For September events, the variability may thus

also be related to the timing of the event within the month.

We found that antecedent conditions such as snow and streamflow before the event can be relevant for the streamflow event response. Jenicek et al. (2016) showed that maximum snow water equivalent in winter (SWE) was influencing low flows in July, especially in catchments above 2000 m elevation and during drier years. Over the summer, the relation in their study became less clear. In our catchments, July is not a low flow period. Still, it appears that winter precipitation has a positive and

spring temperature has a negative effect on June and July warm and dry event responses. This could have two reasons: 1) winter snow may still be present in June and in July at higher elevations (e.g. Engelhardt et al., 2014) (more snow - more snowmelt contribution, higher spring temperatures - less snowmelt) and 2) snowmelt that recharges groundwater in spring contributes to streamflow in July (more snow more recharge, higher spring temperatures less recharge because of saturation). We did not find the positive relation between high spring temperatures and earlier disappearance of snow on the glacier and therefore higher





glacier melt amounts. However, this could relate to MAM temperature anomalies not being a good proxy for snow cover on the glacier.

Prior streamflow has previously been used as an index for carry-over storage in glacierized catchments in Canada to explain summer streamflow variation empirically (Stahl and Moore (2006); and follow up studies Moyer et al. (2016)). Our study confirms that streamflow before the event was found to be particularly important for catchments in Canada, which are in general

larger and may therefore have larger (subsurface) storages. For one of the highly glacierized catchments in Norway, the large part of the variance explained by the antecedent streamflow conditions is likely related to the presence of a lake in this catchment (Nigardsbrevatnet). High antecedent streamflow conditions could reflect filled catchment subsurface storages, recharged by snowmelt, glacier melt and rainfall, or it could mean that warm and dry extreme events (often) occur in summers that are overall exceptionally warm, causing increased glacier melt contributions. In the first explanation, catchment storage contributions

during the event influence the event response (Figure 11), in the second explanation, antecedent conditions have no direct relationship with the event responses. Since the events are distributed over several years, it may not be likely that warm summers are the only explanation of the effect of antecedent conditions on event responses. Studies on groundwater storage in high elevated catchments only emerged recently and mostly focus on non-glacierized alpine systems (Arnoux et al., 2020b; Cochand et al., 2019; Staudinger et al., 2017). Storage contributions to streamflow are also often related to winter or summer low flow

periods, so that little is known about storage contributions to streamflow in summer in glacierized catchments. However, these studies conclude that, contradictory to what long has been thought, there is quite some potential for groundwater storage, especially in quaternary deposits (Arnoux et al., 2020b,a) and there may be some potential for fast and slowly responding storages (Hayashi, 2020; Kobierska et al., 2015). Also, in glacierized catchments there is potential for some groundwater contributions to streamflow in the melt season Mackay et al. (2020); Somers et al. (2019); Somers and McKenzie (2020).

### 5.4   The role of temperature during warm and dry events

The analysis of dry events that were not warm confirmed the importance of high temperatures for a streamflow compensation effect. Without additional melt, or even less than normal melt contributions, streamflow responses to dry periods are on average below normal conditions (ND and CD event responses). This agrees with the findings of **?**, who related streamflow deficits in glacierized catchments to negative temperature anomalies. Compensation levels for ND events may show the relative rainfall

deficits ($100\%$-$C$) as melt is assumed to be normal (Figure 11), but a more detailed look into the rainfall amounts and dynamics of each catchment would be needed to confirm that. From July until September, higher temperatures are favorable to compensate for dry events. However, higher temperatures further downstream may favor high evaporation rates, enhancing the streamflow deficit (Mastrotheodoros et al., 2020). Also, increasing high temperatures or more often occurring relatively extreme high temperatures will not be sustained with higher melt contributions when glaciers retreat. Koboltschnig et al. (2007)

found for example that if the extreme summer of 2003 had happened in 1979 (only changing the glacier outline in the model), the ice melt contribution would have been 400 mm higher. In June, WWD compensation levels were not higher than WD event responses in the Alps and Norway. This may relate to the shorter durations of WWD events and the overall high event responses in June, so that longer events may have a higher positive anomaly.





## 5.5 Temporal variability in event responses

The different hydrological processes behind seemingly similar compensation levels vary over the summer months in the different regions. The precipitation regime determines when most combined warm and dry events occur. For the European Alps, September is, in this case, a very interesting month, as precipitation amounts are relatively low but glacier melt can still provide a buffer to dry periods. For Norway, most events occur in June when snowmelt makes up a large part of the streamflow composition (Engelhardt et al., 2014). In Canada, most events occurred in July, when temperatures can be very high and snow

and glacier melt contributions are important. The rainfall deficits in the different months can be very different, for example, in the Alps they are expected to be much higher during events in July and August compared to September (Figure 2). From June until September melt processes change from snowmelt to glacier melt. The drainage system of the glacier may become more efficient over the summer (e.g. Nienow et al., 1998), possibly explaining the high event responses for some highly glacierized catchments in the European Alps and Canada. Responses in June might be relatively higher than in other months because snow

cover outside the glacier area could increase the area that is sensitive to high temperatures. Still, also in June, responses depend on the relative glacier cover, suggesting that either glacier melt can play a role, or that glacier cover is interrelated with aspect, elevation, or other influences that control the different responses.

Analyzing this unique dataset of long-term streamflow records allowed us to investigate changes over time. Only very few significant trends of $C$ over time were found, mainly in June and September. In June, when streamflow is dominated by

snowmelt, shifts in the regime towards earlier snowmelt may reduce snowmelt amounts (Déry et al., 2009; Ul Islam et al., 2019). For North America, studies also found a decrease in snow amount (e.g. Dyer and Mote, 2006). In Norway, trends in compensation levels tend to be positive and agree with increased melt contributions in recent years (Engelhardt et al., 2014). Positive trends in Norway may also be attributed to increasing trends in snow at higher elevations (Skaugen et al., 2012). In September, negative trends could relate to a reduction in glacier melt contribution because glaciers have retreated to higher

elevations where temperatures in September may not always allow melt. Several studies showed declining streamflow trends in August for Canadian glacierized catchments (Moore et al., 2020; Moyer et al., 2016; Stahl and Moore, 2006). The mainly decreasing trends with time and at the same time increasing occurrence of warm and dry events (Manning et al., 2019; Ridder et al., 2020), suggest that further research is strongly needed to understand how the compensation effect will change and how fast.

## 5.6 Appraisal of an intercontinental multi-catchment analysis

To our knowledge, this is the first study that uses a large set of long-term streamflow observations of glacierized catchments in different regions to carry out an analysis of the compensation effect of glaciers. While this has several advantages, such as analyzing a large sample, it also brought forward some challenges and limitations.

The comparison of different glacierized mountain regions in the world revealed differences in compensation levels between

the three regions, that can be either related to the analyzed sample or regional differences in climate and catchment characteristics. Norway is located at a higher latitude and experiences a strong maritime climate. The melt season may therefore be





shorter, but the glaciers are located at lower elevations than in the Alps and Canada. Also, the results point towards higher importance of snow melt contribution to streamflow in Norway compared to the other two regions during WD events. In general, the compensation levels are lower than in the other two regions, but with high variability, and may relate to the usual

high precipitation amounts (Lussana et al., 2018), and thus high relative deficits. In Canada, catchments are in general much larger, and there are some sub-regional differences in climate, with catchments located close to the west coast, in the eastern part of British Columbia and western part of Alberta, and in the north-west part of Canada. Coastal catchments are therefore more comparable to Norwegian catchments, while the inland catchments may be comparable to catchments in the European Alps. Still, there are quite some differences in minimum glacier elevation between Canada and European Alps, suggesting

differences in climatic conditions. Testing the effect of dry summer conditions by comparing Canadian and European Alps catchments showed some effect, i.e. higher level of compensation in drier catchments, but a more detailed comparison with absolute streamflow amounts and testing event and monthly scale responses would be needed to confirm these findings.

The analyzed sample includes many glacierized catchments. Still, the relative glacier covers are not evenly distributed, which may have caused some bias in the interpretation of regional differences. Most catchments have a relatively low glacier cover

(<15%), especially in Canada. While low glacierized catchments are interesting because they have a more direct practical implication for downstream water users, higher glacierized catchments are useful to isolate better the glacier melt response to warm and dry events. The skewed distribution of glacier covers in the Canadian catchment sample hindered clear conclusions on the effect of reduced summer precipitation when comparing them with catchments from the European Alps, as there were only few Canadian catchments with a glacier cover between 10-50%. In all regions the highest glacierized catchment showed a

clear distinct response. Still, as for the low glacierized catchments, also here there might be a high spatial variability, depending likely more on glacier characteristics and climate variability, but possible also on non-glacierized catchment characteristics such as lakes, groundwater and other catchment storages. However, data from undisturbed highly glacierized catchments are scarce and often not available for extended periods. Zooming in to certain recent events and including more highly glacierized catchments may advance our understanding over the whole range of glacierized catchments responses.

## 5.7 Methodological challenges

Streamflow responses to warm and dry events (and other dry events) were selected to analyze the role of glacier melt in compensating rainfall deficits in the summer months. The streamflow signal, however, likely contained more contributions than only direct melt, e.g. from catchment storages, as we also showed in the analysis of the drivers of the event variability. A (high) rainfall event just before the WD event may potentially still influence the streamflow response during the event and

is then not related to increased snow/glacier melt. Yet, to exclude this effect, detailed knowledge is needed to know the delay between rainfall and streamflow for each catchment (e.g. Stoelzle et al., 2020). Moreover, the effect may depend on the intensity of the rainfall event before the WD event. Shifting the analyzed response window relative to the WD event itself, to exclude the effect of rainfall-runoff processes just before the event and to include the possible delay of glacier meltwater reaching the outlet is not an option, because of conditions after the event that can influence the streamflow response as well. To shorten the

streamflow event responses may be an option, e.g. excluding the first two days. However, the choice of the number of days to





exclude may be arbitrary and vary for each catchment. Moreover, the results showed that WD events are rather limited, and shortening them may result in loss of important information.

Ideally, the glacier melt component itself would be known to assess its compensation effects, instead of analyzing the streamflow signal. We used catchment mean temperature from a grid product and its anomalies to select abnormal melt events.

This does, however, not necessarily relate to absolute glacier melt amounts. A glacier located at lower elevations (and thus higher temperatures) may experience higher melt rates, the same applies to a glacier where the snowline has shifted to higher elevations. Also, at high elevations in the catchment, temperatures may in some cases not reach melt temperatures so that the temperature anomaly does not affect the melt rate. Across regions, temperatures and elevations might be less comparable, as latitude and proximity to the ocean play a role as well. Nevertheless, this study aimed to show how excess melt compensates

for rainfall deficits. Comparing the streamflow responses during these events with the normal streamflow behavior during this time of the year, implicitly considers absolute melt differences between catchments and focuses instead on how much more (relatively) melt is generated.

Summer glacier mass balance observations could be used to obtain information on the amount of glacier melt. For the event scale, however, these observations have a too coarse temporal resolution. Higher temporal resolution measurements of glacier

mass balances would significantly advance our understanding of the glacier buffering role on time scales that matter for water management. Near real-time observations of mass balances may provide some opportunities for that (Landmann et al., 2020). Other data that will be necessary to analyze the different events in more detail are snow data. Snow cover and snow water equivalent (SWE) data may help to estimate the amount of melt that comes from snow instead of glacier melt, to estimate the snow-free glacier area, and to estimate the melt-out date of seasonal snowpack. These variables may be better predictors for

the inter-event variability than the proxies that were used here ($P_{winter}$ and $T_{spring}$).

Part of the limitations mentioned here could theoretically be solved by using glacio-hydrological models, that estimate the different contributions to streamflow and calculate snow amounts (e.g. Engelhardt et al., 2014; Jobst et al., 2018; Stahl et al., 2017). Nonetheless, we need to be careful using such models because there is quite a risk of internal model compensation and on the event scale, this may result in wrong conclusions van Tiel et al. (2020b). Also, modelling one or a few catchments will

not provide the insights that we have shown here, that there is a dependence on relative glacier cover but that there are regional differences and that catchment and climate characteristics play a role too. Thus, the way forward would be to include more detailed observations to extend the empirical analyses and rigorously test models against the observations that we have on the event-scale to use these models for attribution of drivers of the event compensation levels.

## 6   Conclusions

This study provides a unique analysis of the event-scale compensation effect of glaciers to extremely warm and dry periods in the summer months, by analyzing long-term streamflow records of 50 glacierized catchments in Norway, the European Alps and Western Canada. Warm and dry events occur mostly in September in the European Alps, in June in Norway, and in July in Canada. A wide range of compensation levels was found, ranging on average between 50-200% of normal streamflow during



the period of the events. The compensation effect of glaciers depends on several variables, which may all trace back to the
streamflow composition. Above normal (>100%) average event streamflow responses were highest in June and more similar
in July, August, and September. Snow cover in the non-glacierized part of the catchments in June might have enlarged the area
that is sensitive to the relatively high temperatures. The higher the degree of glacierization, the higher the relative streamflow
amounts during an event. This signal was most apparent for catchments in the European Alps and Canada. The glacieriza-
tion levels where the average compensation level switches from below (no full compensation) to above normal streamflow
(compensation or overcompensation) were found to be lower for catchments in Canada ( 5% $gc$) compared to the European
Alps (10-15% $gc$), possibly related to differences in precipitation amounts in summer (dry in Canada, wet in European Alps).
Besides these variations in compensation levels related to timing in the summer, the degree of glacierization, and regional
differences in climate and catchment characteristics, the event compensation levels for individual catchments and months also
showed a large range, with levels below and above normal conditions. Antecedent conditions that relate to precipitation in win-
ter and streamflow anomalies 30 days before the event can explain part of this variability, suggesting that catchment storages,
in the form of snow, water stored in the glacier, groundwater and lakes, are important too for the effect of excess melt on the
relative streamflow response.

Understanding these different mechanisms behind the event responses and glacier compensation effect is necessary as dry
periods occur in different parts of the year in different regions, and the responses are thus driven by different processes. These
insights will also be crucial to estimate these systems' sensitivities in a changing climate with increasing temperatures and
changing precipitation patterns and increased risk of these combined warm and dry events. Furthermore, the event-scale obser-
vations in the different summer months also have implications for our understanding of longer-term droughts and heatwaves
and how they will affect the development of hydrological drought over the summer.

*Data availability.* Streamflow data from Austria are openly available from the eHYD database at https://ehyd.gv.at/. Swiss streamflow data
were provided by the Swiss Federal Office for the Environment (FOEN) and available there on request. Canadian streamflow data is openly
available via the HYDAT database (https://wateroffice.ec.gc.ca/mainmenu/tools_and_downloads_index_e.html). Streamflow data for Nor-
way was provided by the Norwegian Water Resources and Energy Directorate (NVE) and available there on request. Climate data were made
available by Zentralanstalt für Meteorologie und Geodynamik (ZAMG) and Meteoswiss, for Austria and Switzerland, respectively and are
available on request from these institutions. For Norway (seNorge2) and Canada (PNWAmet), climate data are available according to the
given references in the text. Glacier outline data and mass balance data are available according to the given references (WGMS). The source
of the background images in Figure 1 is Esri, DigitalGlobe, GeoEye, Earthstart Geographics, CNES/Airbus DS, USDA, USGS, AeroGRID,
IGN and the GIS User Community.

*Author contributions.* MVT and KS conceived and designed the study. MVT performed the analyses and wrote the original manuscript. KS
and AVL provided feedback on the analyses. All authors reviewed and edited the final article.



555 *Competing interests.* The authors declare that they have no conflict of interest

*Acknowledgements.* MVT and KS were funded by the DFG project STA632/4-1 Tracing trends and changes of drought in hydrosystems.



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
