# Peer review of "Hydrological response to warm and dry weather: do glaciers compensate?"

_Hydrology and Earth System Sciences, 2021_

## Referee Comment (RC1)

**Review of the manuscript „Hydrological response to warm and dry weather: do glaciers compensate?" submitted by Marit Van Tiel, Anne F. Van Loon, Jan Seibert, and Kerstin Stahl to Hydrology and Earth System Sciences (HESS)**

**Review by Mauro Fischer (Glaciologist at the Institute of Geography, University of Bern, Switzerland, mauro.fischer@giub.unibe.ch)**

**General comments**

I want to thank the authors for their noticeably large amount of work for this very nice study and valuable scientific contribution. In my opinion, the study contains novel and highly interesting findings, the presented results seem solid overall (except maybe for how the authors calculated catchment-wide mass balance, see comments below) and are of high interest to the scientific community. In general, the manuscript is well written and very nicely illustrated (great figures!). I think that this work clearly deserves to get published in Hydrology and Earth System Sciences. The authors will need to put some effort into correction and improvement of their manuscript. In my opinion, there are still (even if only few) important major and a number of minor issues which need to be addressed, corrected, clarified, and implemented prior to publication. I guess that the majority of the specific comments listed below are easy to implement, whereas some general or specific comments will maybe need some additional work and time. I hope that my work for this review will help improving the paper, and I encourage the authors to implement and reply to all my comments as far as possible. Thanks a lot, it was a pleasure to review your study, congrats, all the best and kind regards.

*List of some general comments:*

- Abstract: very well written and very good overall! See some specific comments below. As you analyzed glacier-fed streamflow responses to WD in long-term streamflow observations (>50 years), in my opinion it would be beneficial for people interested in your paper to add some information about your observation periods for Canada, Norway, and the European Alps directly in the abstract as well…

- Introduction: I like the structure/storyline and content of the introduction, maybe it is a bit (too) long and could be shortened without losing important information and content? Sometimes I wondered whether the difference between "the buffering effect of glaciers" and "the compensation effect of glaciers" is always clear (or if there is any big difference at all…). Maybe you could check that point while going through the introduction (or the whole manuscript) again? Moreover, please find a number of specific comments which hopefully help to improve specific parts of the introduction below…

- Data: I have, unfortunately, some concerns about how you deal with and use measured mass balance data in your study; you cannot just take a median of measured glacier mass balance data, you have to weight the measured data with glacier area, i.e. calculate area-weighted average mass balances, otherwise you might get wrong results. This is relevant when it comes to compare or correlate mass balance with levels of compensation $C$, I clearly think you need to check and clarify that; moreover, it does

not really make sense to me to just use "country-wide" mass balance data for individual catchments you analyze, there are better ways to extrapolate measured mass balance data to individual catchments, especially in areas with a high density of measured mass balance data like Norway, Switzerland, Austria. Please see also specific comments thereupon below.

- Results: In section 4.2 you only refer to WD (or WWD) events, right (CD, and ND events are excluded)? Maybe make that clearer in the text (e.g. by changing the title of the section to "4.2 Glacier compensation during WD events in different catchments"); it was not very much clear to me why the chosen catchments and years for individual regions were selected for Figure 3 (just an example or do these three years and catchments reflect some of the "general observations, trends, and *C* values over the whole observation period"?)…

- Results: Section 4.4 and figure 9: As I understood how you calculate catchment-wide mass balance or take into account measured mass balance data to correlate it with levels of compensation *C*, in my opinion this is not quite correct (see comments above and specific comments thereupon below), maybe by correctly calculate catchment-wide mass balance your resulting correlations of mass balance with levels of compensation would quite change and show another relation between these variables?!...

- Results: Still section 4.4 and figure 10: To be honest, I didn't really understand how you calculated correlations between changes in *C* and glacier changes over time (mostly retreat for your period of observation, but with intermittend phases of readvance, for instance for Norwegian glaciers in the 1990s!), this is neither very clear from the methods section nor very clear in the results section… what data sets did you take into account to check if glaciers in the analyzed catchments did change in area and volume over your observation period? – this aspect, relating also to figure 10, is not very clear…

- In the whole manuscript: I think it would make more sense to use "southwestern Norway" instead of "Norway" and "western Canada" instead of "Canada" in the entire manuscript (these countries are so large and you analyze only catchments in specific regions)…

**Specific comments and technical corrections**

To facilitate the author's correction of the manuscript I combined specific comments and technical corrections (including language or spelling and comprehensibility issues). Sometimes comments contain both specific comments and technical corrections, sometimes just one or the other. I would ask the authors to implement my comments and suggestions as far as possible.

***Title***

*Personally, I am also for short and concise, attractive titles (and I like yours), but maybe it would be good to add short information about your period of observation and study areas in the title? Up to you…*

***Abstract***

Lns 11f: "C was, in general, higher than 100% for catchments with a higher relative glacier cover" → *is it possible to give a number of the (average) relative glacier cover (XY% of the catchment glacierized over the observation period) needed to result in a C ≥ 100%? Would be quite interesting…*

Ln14: "…which indicated the…" → *rather use present than past here as statement is still valid?*

Ln16: "…such as the snow fallen…" → *"…such as the amount of snow fallen…"*

Ln17: "Overall, these results suggest…" → *"Overall, our results suggest…"*

Lns17ff: *I know that you have to summarize and make short statements in the abstract, but, in my opinion, the last two sentences of the abstract could be a bit clearer and more precise (how do glaciers not compensate straightforwardly? What are "the different streamflow contributions and their variations" you refer to here?)*

***Introduction***

Ln 22: "…alter the input of hydrological systems…" → *"…alter the water input of hydrological systems…"?*

Ln 30: "In high mountain regions, where snow and ice are present, these snow and ice storages…" → *repetition of "snow and ice", maybe rephrase for smoother readability?*

Ln 31: "…, because of temperature-driven water supply." → *why not directly write what you refer to here? i.e. enhanced snow and ice melt due to high temperatures?...*

Ln 33: "…of catchments(e.g. Bakke…" → *insert space between "catchments" and "("*

Ln 34: "…also known to be a buffer against…" → *better:"…also known to act as a buffer against…"?*

Ln 34: "…these storages can also be themselves be depleted…" → *one "be" too much →  "…these storages can also be depleted themselves…"*

Lns 42f: "Hence, groundwater and snow storages might not always be a perfect buffer during warm and dry periods." → *Please bee a bit clearer and more precise here, I guess what you want to say here is something like "Hence, groundwater only has a limited buffering capacity*

*(in terms of runoff, provides only baseflow), while the buffering capacity of snow is higher (in terms of additional runoff) but temporally limited (if all snow has melted, there is no buffer anymore).*

Lns 44f: "…a favorable buffer during such periods…" → *"…a favorable buffer during warm and dry periods…"*

Ln 46: "…in the year…" → *"…throughout the year…"?*

Lns 48f: "During such extreme drought years…" → *why "such"?*

Ln 49: "…runoff from glacier areas…" → *"…runoff from melting glaciers…"?*

Ln 49: "…55-100% of summer runoff…" → *"…55-100% to summer runoff…"?*

Ln 55: "…from other streamflow contributions such as snowmelt." → *maybe add some additional important streamflow contributing processes (in high mountain areas) here → "…from other streamflow contributions such as snowmelt, surface runoff, interflow, groundwater flow or melting permafrost."?!?!*

Ln 58: "…during such extreme warm and dry periods." → *why "such"?*

Lns 60ff: "During warm and dry years, glaciers can provide more meltwater to streamflow, and during cold and wet years they generate less meltwater so that altogether the interannual streamflow variability is relatively low." → *do the studies you refer to here give a "lower threshold ratio of glacierization" for individual catchments (i.e. minimum percentage of glacier-covered area in a catchment) from which the dampening effect of glaciers on interannual streamflow variability applies? – would be interesting added value here…*

Ln 64: "The result of the balancing between melt and precipitation is assumed…" → *In order to be clearer, I'd rather write "Whether the amount of runoff from melting glaciers has a dampening effect on the interannual streamflow variability or not is assumed…"*

Lns 66f: "…but also other climate and catchment characteristics appear to influence the streamflow sensitivity to climatic anomalies…" → *other climate and catchment characteristics like what? – can you give some examples mentioned in the studies you refer to here?*

Ln 69: "…mountain glaciers have been retreating…" → *"…mountain glaciers around the globe have been retreating…*

Ln 72: "…provided an additional source of water in the summer compared to the seasonally delayed contribution." → *why "COMPARED TO the seasonally delayed contribution"? rephrase in order to be clearer…*

Ln 73: "However, this source will not be sustained" → *"However, additional meltwater from shrinking glaciers will not be sustained forever."*

Ln 74: "…some regions…" → *"…some glacierized mountain areas…"*

Ln 78: "…differs in different catchments…" → *"…varies for different catchments…"*

Ln 80: "…than low glacierized catchments." → *"…than catchments with low relative glacier cover".*

Lns 80ff: "However, this contribution may reach a maximum, as the higher glacierized catchments are generally located at higher elevations that receive more orographic precipitation amounts…" → *can you please check that statement? In my opinion, it is not primarily the increased amounts of precipitation that causes a maximum value of relative catchment glacier cover in terms of increased streamflow contribution by glacier melt, but rather air temperature and climatic conditions in highly glacierized catchments, isn't it?*

Ln 83: "…is generally assumed to be highest in August and September, …" → *of course, this is only true for mountain glaciers in the northern hemisphere with a mass balance regime similar to the one in the Alps… I am sure you are aware of that but it might be good to be more precise here… (as your study is interesting to people from all over the world ;-))*

Ln 85: "…relative contribution is, …" → *"…relative glacier melt contribution is, …"*

Ln 96: "This scale enables…" → *"The chosen time scale enables…"?*

Ln 97: "Moreover, this scale…" → *"Moreover, this time scale…"*

Lns 99f: "…we analyzed observed hydrological responses to WD events for catchments with varying glacier cover in Norway, Canada, Switzerland and Austria." → *I recommend to add some information about the observation periods / temporal time frame of your study here.*

***Data and hydroclimatology of selected glacierized catchments***

Ln 108: "…based on length of the time series (long records), …" → *"…based on the length of available data time series (long records), …"*

Ln 109: "…missing values…" → *"…data gaps…"*

Ln 109: "…(including low and high glacierized catchments)…" → *…"(including catchments of low to high relative glacier cover)…"*

Lns 109f: "A few of these catchments are nested." → *What does this exactly mean here? – Can you maybe briefly explain in key words in parentheses after the sentence?*

Ln 110: "…derived from gridded data products…" → *"…derived from gridded reanalysis data…"?*

Ln 115: "…for the different catchments differed, between 50 and 68 years." → *"…for the different catchments varied between 50 and 68 years."*

Ln 118: "…but the selection of events…" → *"…but the selection of WD events…"?*

Ln 119: "…occurred mostly in the winter months." → *"…coincide mostly with the winter months."*

Lns 122f: "Therefore, regional average mass balance time series were calculated from all available mass balance observations per year per country (Austria, Switzerland, Norway and Canada) by taking the median." → *If I understood your approach right, then, as a glaciologist, I have some concerns here:*
*1) Taking just an average or median value from measured mass balance data of individual glaciers is not really correct, you need to calculate area-weighted average mass balance values (as for instance large valley glaciers have their ablation area at lower elevation (therefore more melt, therefore more negative mass balance) compared to smaller mountain glaciers having their terminus at higher elevation (therefore less melt, therefore less negative mass balance)); i.e. you have to multiply all mass balance data you take into account by glacier area, then calculate the sum of these values for all glaciers, and finally divide that sum by the summed up glacier areas*
*2) Even though for instance Switzerland and Austria are small countries, measured in-situ mass balance sometimes comes from glaciers with significantly different regional climatic conditions (the same must apply for Canada, for Norway your catchment sample lies in a more or less similar climatic zone I guess)… So my point here is that, in my opinion, it does not make sense to calculate average mass balance values "by country" for the catchments you analyze! For example, Schaefli et al. (2019, The role of glacier retreat for Swiss hydropower production, Renewable Energy) calculated area-weighted mass balance data for all glacier-covered catchments with hydropower plants using data by Fischer et al. (2015, Surface elevation and mass changes for all Swiss glaciers 1980-2010, The Cryosphere). I think you should find a way to reasonably derive catchment-wide mean annual mass balance data from existing measured mass balance data and extrapolation techniques (for the latter see for instance Huss (2012), Extrapolating glacier mass balance to the mountain-range scale: the European Alps 1900–2100, The Cryosphere); you might also work with Huss and Hock (2015, A new model for global glacier change and sea-level rise, Frontiers in Earth Science, there, past and future glacier mass balance data is available (per glacier!) from the Global Glacier Evolution Model (GloGEM)). Whatever data you use or method you apply to extrapolate (measured) mass balance data to glaciers of your analyzed catchments, do not forget that you have to calculate area-weighted values again if you have to work with "catchment-wide" mass balance data for your analyses. If you want you can also contact me personally (mauro.fischer@giub.unibe.ch) and I would be happy to discuss that with you and try to help you there.*

Ln 128: "…(GI4, 2015)…" → *I think you need to add the correct reference for the Austrian glacier inventory used here.*

Ln 130: "…Randolph glacier inventory (RGI) for Canada…" → *maybe add the version of the RGI you worked with (as there were a lot of updates from the first to the latest RGI version)…*

Ln 128: "…ranging in size from…" → *rather put the "in size" at the end of the sentence (after the numbers)*

Ln 174: *I would delete the "(not shown)" here… but would it maybe make sense to add a table with catchment name, location, size, mean elevation, mean annual temperature, mean annual precipitation and resulting levels of compensation to the Supplementary Materials? – This would add some relevant detail for people interested in catchments you looked at…*

***Methods***

Ln 181: "WD spells…" → *"WD periods…"?*

Ln 188: "…e.g. van Loon, 2015))" → *Delete one of the parentheses there*

Ln 202: "Once the periods…" → *"Once the WD (or other dry event) periods…"?*

Lns 202ff: "It was assumed that in these high elevation catchments there is an immediate response to the WD events (or the other dry events) and streamflow data of the dates exactly corresponding to the events were selected." → *can you add some information on why (from a process understanding point of view) it is ok to work with that assumption? I argue that you're right with this assumption but giving some rationale here would be good I think.*

Ln 208: "…Qn…" → *"…$Q_n$…", also everywhere else in the text?*

Ln 216: "*C* of the events…" → *"Resulting C values of the events…"?*

Ln 219: "…general trend of streamflow during the event." → *"…general trend of streamflow behavior during the event."?*

Lns 227f: "…the importance of different event drivers." → *as for example?*

Lns 228f: "glacier cover" → *"relative glacier cover" (twice)*

Ln 231: "…at least 8 or more events…" → *what is your rationale behind this threshold?*

***Results***

Ln 235: "…(Table 3." → *"…(Table 3)."*

Lns 240f: "…and different numbers of catchments per region." → *I would also argue that they are not comparable due to different (hydro-)climatological settings of the individual catchments!?*

Ln 242: "…occurred in the year 1982…" → *"…occurred in 1982…"*

Ln 250: *I would delete both commas there…*

Ln 257: "…the importance of glacier cover…" → *"…the importance of relative glacier cover…"*

Ln 260: *add a comma before "too."*

Ln 262: "…for the two lower *gc* catchments…" → *"…for the two catchments with lower gc…"*

Ln 263: *I would delete the "pushed"*

Ln 265: "…compared to the regime…" → *"compared to the long-term daily streamflow regime…"*

Ln 270: "…part of the variance of $C$…" → *"…part of the variance in $C$…"*

Ln 275: "…below the normal regime…" → *"…below the long-term daily streamflow regime…"*

Ln 279: "The spread in responses…" → *"The variability in streamflow responses…"*

Ln 287: "The sreamflow responses of WWD and WD events…" → *"The streamflow responses to WWD and WD events…"*

Ln 291: "…were below the normal regime…" → *"…were below the normal streamflow regime…"?*

Lns 295f: "Different rainfall amounts…"; "…high rain amounts in summer…"; "…low rain amounts in summer…" → *I guess you refer to the climatological statistics here (how much rain falls on average in one region in summer, cf. figure 2)… I would write that somehow in this sentence in order to be clear…*

Ln 297: *I think you can delete the comma there…*

Ln 298: "Having less rainfall…" → *"Less rainfall…"?*

Lns 300f: "Also, the relative glacier covers of the Canadian and Alps catchments are more complementary than comparable in this sample of catchments." → *I believe to see your point here, but doesn't make this statement figure 7 and your rationale/descriptions here about the influence of the amount of average summer rainfall on the levels of compensation a bit obsolete?...*

Ln 304: "…show these sometimes rather large ranges. To explain these ranges,…" → *"…show this sometimes large variability in catchment-wide $C$. To explain this variability,…"*

Ln 306: "…variance of $C$…" → *"…variance in $C$…"*

Lns 323f: "…when $MB_{sum}$ is larger (more negative)." → *"…when $MB_{sum}$ is more negative." →* *as I wrote above, try to avoid speaking of larger/higher and smaller/lower mass balance, this is always confusing (use "more negative" and "more positive")…*

Lns 328f: "Most significant trends were found in June (Canada) and September (European Alps), which were all negative. Norwegian catchments showed mostly positive trends, except in September." → *ok, what does that mean? Can you maybe relate these correlations to observed glacier changes in the analyzed catchments? For Switzerland, see for instance M. Fischer et al. 2014 (The new Swiss glacier Inventory SGI2010, Arcitc, Antarctic and Alpine Research, section Study Region), for Austria, see for instance A. Fischer et al. 2015 (Tracing glacier changes in Austria from the Little Ice Age to the present using a lidar-based high-resolution glacier inventory in Austria, The Cryosphere), for Norway see for instance*

*Winsvold et al. 2014 (Glacier area and length changes in Norway from repeat inventories, The Cryosphere), for western Canada see for instance Bolch et al. 2010 (Landsat-based inventory of glaciers in western Canada, 1985-2005, Remote sensing of Environment)…*

**Discussion**

Ln 334: *why "such"?*

Ln 340: "…daily regime…" → *"…daily streamflow regime…"?*

Ln 341: "…during such an event…" → *"…during a WD event…"?*

Ln 346: "…as in these rapidly changing systems…" → *"…as it concerns rapidly changing systems…"?*

Ln 346: "…daily regime…" → *"…daily streamflow regime…"?*

Ln 350: "…high summer." → *"…midsummer."?*

Ln 351: "…such events…" → *"…WD events…"?*

Ln 351: "…in the summer shoulder season…" → *"…in midsummer…"?*

Ln 354: "Glacier cover…" → *"Relative glacier cover…"?*

Ln 360: "…for events…" → *"…for WD events…"?*

Lns 366f: "But Figure 4 shows that there are exceptions." → *I guess you refer to catchments and C values in Canada here? – Thus, exceptions concern catchments in drier average summer climates (cf. Figure 2) with already low relative glacier cover? → be more concrete/precise here…*

Ln 370: *Is there a comma lacking after the parentheses?*

Ln 371: *Is there a comma lacking after "conditions"?*

Ln 372: "…also make it difficult to answer when glaciers compensate because it depends on the situation." → *a bit vague and not very clear to me, can you rephrase in order to be more precise here?*

Ln 373: "…these variations…" → *"…this variability…"?*

Ln 376: *Why do you refer to Figure 11 here? – Shouldn't you refer to figure 8?*

Ln 382: "…was influencing…" → *"…is influencing…"?*

Ln 383: "…above 2000 m elevation…" → *"…above 2000 m a.s.l…"?*

Ln 384: "…became less clear…" → *"…is less clear…"*?

Ln 388: "(more snow more recharge, higher spring temperatures less recharge because of saturation)" → *"(more snow - more recharge, higher spring temperatures - less recharge because of saturation)", be consistent with Lns 386, 387…*

Ln 390: "…MAM temperature anomalies…" → *"…spring temperature anomalies…"? or is it clear for anyone that "MAM" refers to March, April, May?*

Ln 392: *Can you shortly explain in parentheses what "carry-over storage" exactly means?*

Ln 394: "…was found to be…" → *"…is…"*

Ln 399: "In the first explanation…" → *"In the former case…"?*

Ln 400: *Why do you refer to Figure 11 here? – Shouldn't you refer to figure 8?*

Ln 400: "…in the second explanation…" → *"…in the latter case…"?*

Lns 402f: "…in high elevated catchments…" → *"…in high mountain catchments…"?*

Ln 409: "…Mackay et al. (2020); Somers et al. (2019); Somers and McKenzie (2020)…" → *"(Mackay et al., 2020; Somers et al., 2019; Somers and McKenzie, 2020)"*

Ln 413: *There is a reference lacking here I guess…*

Lns 418f: "Also, increasing high temperatures or more often occurring relatively extreme temperatures will not be sustained with higher melt contributions when glaciers retreat." → *this is only true for time periods after "peak water" of individual catchments, I am sure you know that, but maybe you could include that in some way in your statement here…*

Ln 421: "…400 mm higher." → *I am not sure how I can interpret this value, is it per $m^2$ (for the entire catchment)? Add some information here…*

Lns 422f: "This may relate to the shorter durations of WWD events and the overall high event responses in June…" → *add "…due to enhanced contribution from snowmelt compared to the later summer season…" in parentheses after "in June"?*

Ln 432: "…may become…" → *"…becomes…" (as, at least for temperate glaciers, this is always the case in summer)*

Ln 438: "…changes over time." → *"…changes in streamflow response to WD events over time."?*

Ln 439: "…trends of *C* over time…" → *"…trends in C over time…"*

Ln 447: "…decreasing trends with time…" → *"…decreasing trends in C with time…"?*

Ln 448: *I would delete the comma here; add "in future" after "…will change…"?*

Ln 445: *I would delete "in the world"*

Ln 446: *I think here it becomes obvious that it would make sense to talk/write about "southwestern Norway" and "western Canada" in the entire manuscript (because you clearly make a statement about southwestern Norway and not the entire glacierized catchments of Norway here… (see also general comment thereupon above)…*

Ln 464: "…Canada and European Alps…" → *"…Canada and the European Alps…"*

Lns 484f: "…and is then not related…" → *"…which is then not related…" (as you refer to streamflow and not to high precipitation amounts before a WD event here…*

Lns 489f: "To shorten the streamflow event responses…" → *"To shorten the analyzed time frame of streamflow responses to WD events…"*

Ln 492: "…shortening them…" → *"…shortening the analyzed time frame…"*

Ln 494: "…grid product…" → *"…gridded product…"?*

Lns 495f: "…(and thus higher temperatures)…" → *"…(with thus higher temperatures)…"?*

Ln 497: "…melt temperatures…" → *"…melting conditions…"*

Ln 500: "…during these events…" → *"…during WD events…"*

Lns 500f: "…during this time of the year…" → *"…during the summer months…"*

Lns 504f: "Higher temporal resolution measurements of glacier mass balances…" → *"Measurements of glacier mass balances with higher temporal resolution…"*

Ln 509: "…melt-out date of…" → *"…disappearance date of…"?*

Ln 511: *I would delete the comma here*

Ln 513: "…there is quite a risk of internal model compensation…" → *can you add some information here (maybe in parentheses)? What does that actually mean? What exactly do you refer to here?*

Ln 514: "…van Tiel et al.(2020b)… " → *"…(van Tiel et al., 2020b)… "*

Ln 516: *add a comma before "too."*

**Conclusions**

Ln 521: "…in Norway…" → *"…in southwestern Norway…"*

Ln 532: "Besides these variations…" → *"Besides the variability…"?*

Ln 534: "…with levels…" → "…with compensation levels…"

Ln 535: "…streamflow anomalies 30 days before the event…" → "…streamflow anomalies during 30 days before the event…"

Ln 538: "Understanding these different mechanisms…" → "Understanding the different mechanisms…"

Ln 541: Delete "these"

Ln 543: I would add "in high mountain catchments" before "over the summer"

**Figures**

Figure 1: Maybe add scales to a), b), c). I would write "Norway" (in green), "European Alps" (in blue), and "Western Canada" (in pink) at the lower right of the figure 1d). Moreover, can you add a more detailed legend for the circle sizes which signify the relative glacier cover of a catchment (i.e. how much relative glacier cover do the different circle sizes in 1d) represent?). Moreover, would it possibly make sense to additionally draw areas with "similar hydroclimatological regimes" (cf. chapter 2.3) in Fig. 1? – Maybe this would add valuable graphical information…

Figure 2: Looking at figure 2 I cannot really understand why two different graphs for Norway, four different graphs for Canada, and three different graphs for the European Alps. Can you please add something on that in the figure caption? You explain it in chapter 2.3 but I think to add some information thereupon in key words would be good for the figure caption.

Figure 3: You have to add a bit of information in the figure caption there: Write what exactly the i) black line, ii) blue line, and iii) the red dots signify (I know it's written in abbreviations at the bottom of the graph but it would be helpful to have that information (as text) in the figure caption as well). The same for the relative glacier cover (bold black number in the upper right corners), write it as text in the figure caption as well to be clear. Can you please also add somewhere in the manuscript (text or figures) based on what you chose to show the selected results for some selected catchments and selected individual years for the three regions (shown in Figure 3)? What was your rationale behind that? And maybe also which catchments (names, location) are shown?

Figure 4: Figure caption: Write "Mean catchment level of compensation…" as in the figure caption of Figure 6!?; "…indicate the mean response…" → "…indicate the mean streamflow response…"?

Figure 5: Figure caption: "…during the WD in different regions…" → "…during the WD events in different regions…"

Figure 6: Figure caption: maybe add "…in different regions (rows) and months (colums) at the end of the first sentence to be consistent with figures 4 & 5?!; "If all dry periods occurred…" → "If all dry periods occurred at the same time…"?

Figure 7: *Figure caption:* "…that have low summer rain…" → *"…with low average summer rain amounts…"*; "…and high summer rain amounts…" → *"…and high average summer rain amounts…"; maybe add again information about what the pink and blue bars mean (see e.g. figure caption of figure 5)…*

Figure 8: *Figure caption:* "Explained variance of C…" → *"Explained variance in C…"; moreover, neither from the figure nor from the text it is evident how the thresholds to separate individual classes of relative glacier cover were chosen… can you add some information on that please?*

Figure 9: *Figure caption: Delete the "9" at the beginning of the figure caption; you would have to write out the used abbreviations for annual, winter and summer mass balance somewhere (I would argue better in the text than here in the figure caption, for $MB_{win}$ you did it in the text);* "Colours and symbols indicate the three regions (as in other figures)." → *why not be concrete and directly write colours of symbols together with corresponding regions here; can you explain the "$r_s$" (y-axes)? I would just add ($r_s$) in the figure caption in parentheses after "Spearman rank correlation"…*

Figure 10: *Figure caption: I would again write colours of symbols together with corresponding regions here; moreover, it is not really clear to me how time trends of C were related to time trends of relative glacier cover here (see general comments thereupon above…), I would need some more information in the figure caption to know how to exactly interpret this figure…*

Figure 11: *Figure caption: I would delete the "the deficit" at the end of the figure caption; am I right that this conceptual figure is only valid for the situation during summer season? – I would add that in the figure caption…; would it not be less misleading if you write "Q (high gc)" and "Q (low gc)" in the figure rather than "Q (>>gc)" and "Q (<<gc)"?...*

**Tables**

Table 1: *Add full stop at the end of the table caption.*

Table 2: Table caption: "…level of compensation. (*C*)" → *"…level of compensation (C)."*; "Higher elevated glaciers:…" → *"Glaciers at higher elevation:…"*; "Percentile of mean temperature in spring (MAM)" → *which percentile?*; "Percentile of sum of precipitation in winter (DJF)" → *which percentile?*; "The streamflow percentile of the 30 days before the WD event" → *which percentile?*; "Higher $MB_{sum}$…" → *I would delete that because "higher" means more positive and here you mean more negative (it is always easier to talk about "more negative" or "more positive" glacier mass balances in order to avoid misunderstandings when using "higher" or "lower" mass balances!), moreover you use the abbreviations $MB_{sum}$ and $MB_{win}$ but don't explain them (I can easily guess what it means but someone else not too familiar with glaciers might not at first glance); please see also my comments above about how you calculated "regional mass balances", I think if you calculate catchment-wide mass balance or mass balance anomalies in a more appropriate/more correct way you can use these numbers much better for a more realistic interpretation of "C" with the help of mass balance data…*

Table 3: *Table caption:* "in brackets" → *"in parentheses"; Add full stop at the end of the table caption;* "Years most events" → *"Years with most events"; in my opinion it would be good to add the actual observation periods (e.g. 19XY-19XY) in parentheses after "Alps", "Norway" and "Canada"!*

Table 4: *Table caption:* "The numbers in between brackets…" → *"The numbers in parentheses…"; in my opinion it would be good to add the actual observation periods (e.g. 19XY-19XY) in parentheses after "Alps", "Norway" and "Canada"!*

---

## Referee Comment (RC2)

**Reviewer reply on AC1-hess-2021-44 by Mauro Fischer, University of Bern**

Dear Marit and co-authors of the manuscript "Hydrological response to warm and dry weather: do glaciers compensate?", submitted to Hydrology and Earth System Sciences. Many thanks for your reply on my review. – Please see my answers to your comments and author replies below.

*We appreciate the reviewer's positive evaluation. Thank you for the fast review with useful and detailed comments, which will improve the manuscript. To facilitate some discussion, we will respond below to the general comments and provide a detailed reply to the specific/technical comments with the actual revision.*

**Reply:** You're very welcome. Thank you for your nice and interesting work. I would very much appreciate if you could also provide a document with point-by-point comments on how you implemented all the specific and technical comments (also regarding figures and tables) of my review in the end to the public discussion.

- *Abstract: the reviewer suggests to add the observation periods for the different regions in the abstract. We agree that this is a good idea. However, the different catchments have all different observation periods. For the Alps, streamflow and meteorological data overlap for the period 1961-2015/2016. Still, for Norway and Canada, individual catchments have a different observation period (but all at least 50 years), and therefore it is not easy to summarize that in the abstract. Nonetheless, we could indicate the range of years that were analyzed (1945-2016).*

**Reply:** I see your difficulties here regarding different observation periods for different regions. – I think adding/indicating a range of the analyzed period in the abstract is fine.

- *Introduction: When revising the manuscript, we will try to shorten the introduction - thank you for pointing that out. We will also carefully check again the use of the terms buffering and compensating. In our opinion, these concepts mean something different but are in the literature often used as synonyms when describing glaciers and droughts. We addressed this in the discussion but suggest that we will clarify this difference already in the introduction in the revised version. In our view, buffering means 'something that helps protect from harm', i.e. it is usually a passive process but could also be an active process. In our context, glaciers act as a buffer to (meteorological) droughts, because they still provide water during periods when rainfall input is low, thereby preventing a severe impact on water availability, especially in relative terms. In this respect, glaciers are always a buffer, regardless of the situation, because their water supply (on the short- term) is dependent on temperature and radiation rather than rainfall. 'To compensate', in our view, means 'to provide something to reduce the effect of something that has been lost or damaged', i.e. it is an active process. In our context, it means that excess (more than average) glacier melt can reduce the effect of a rainfall deficit on streamflow.*

  *In the case of buffering, the effects of a rainfall deficit on streamflow can still be present, but streamflow will not be affected as much as without a buffer present, because part of the streamflow comes from groundwater/snowmelt/glacier melt. In case of compensation, excess glacier melt can compensate for the rainfall deficit, and streamflow can be close to average or above normal levels. Since glaciers do not only buffer but also compensate during warm and dry events, we argue (in the discussion) that such a distinction should be made and that this approach helps to quantify this specific role of glaciers. We would appreciate to hear from the reviewer whether we here were able to clarify our argumentation.*

**Reply:** Many thanks for your arguments and clarifications here. Indeed, I now much better understand how you differentiate between "buffer" and "compensation". If you integrate these arguments – as you write – in the introduction and discussion, I think this should be all fine and much clearer for the readers of your paper.

- *The reviewer has some concerns about how we used measured mass balance data in our analyses. One concern relates to the way we averaged different mass balance observations to have country-wide mass balance time series, and the other concern relates to the use of this country-wide mass balance instead of having mass balance time series for each individual catchment.*

  *The aim of this analysis was to investigate how event-scale level of compensation relates to variations in annual, winter and summer balances, keeping in mind the different time scales analyzed (one event versus integrated response over the year, or season). Since only a few mass balance measurements are available, and only a few of them have long- term observations, we decided to combine them in one country-wide mass balance time series. This time series was then used to test correlations with C of each catchment in that respective country. We do acknowledge the reviewers suggestion on using area- weighted average mass balance on country scale and suggest that we will recalculate region-specific mass balance time series using the same approach. We tested the effect of using area-weighted average mass balances (for Austria and for Switzerland) for the correlations between C and mass balances in the European Alps (see Figure below), and observed some small differences. We suggest that we will use the area-weighted average approach in the revised manuscript.*

**Reply:** I appreciate a lot that you can see my concerns about how you calculated and applied measured mass balance data in the first submitted version of the manuscript. Your figure below shows that, even if differences are at first glance mostly small (except maybe for the column September), country-wide seasonal mass balances calculated using area-weighted data produces different results than by "just taking a median of measured mass balances". I totally agree that you will use area-weighted data in the revised manuscript.

  *Regarding the use of these regional/country averages mass balance time series, we agree that this is not the best approach. Still, we are rather limited because of limited observations. The ideal option would be to have long-term mass balance time series available for all glaciers in each catchment. In reality we are far from that situation. The reviewer suggests to not average mass balance time series per country because of significant regional climatic conditions. Using geodetic mass balance data, or mass balance data calculated from a model, as done in the studies referred to by the reviewer, is not really an option here, as we need annual variations and would like to base this study solely on observations rather than mixing observations and simulations. We would thus need to find a way to extrapolate the few observations that exist, to all the other glaciers in the studied catchments. The study of Huss (2012) nicely illustrates different options to do that, i.e. arithmetic average, using glacier hypsometry or multiple regression. Although Huss did such an analysis for the European Alps, doing something similar for southwestern Norway and western Canada is beyond the scope of this study. The only option left would be to find out which catchments have a measured glacier in their boundaries that has long-enough time series to do a correlation analyses with the level of compensation. Area-weighted averages will not need to be used then, as it is unlikely that more than one glacier in the same catchment is measured. This will reduce the number of catchments analysed in this part of the study significantly, but may represent better the relation between mass balances and level of compensation and will be worth testing.*

**Reply:** I am totally aware of the fact that long-term measured mass balance data is lacking for a lot of catchments you analyzed. Still, at least for areas with comparatively (spatially) very dense and rather long-term measured mass balance time-series like Switzerland or Austria (or maybe even southwestern Norway?!), I think it would be worth taking only mass balance data from glaciers with "comparable regional climate conditions" for analyses of your catchments (e.g. differentiate between catchments of the northern slopes of the alps, of the inner (high) alpine regions (there also between west and east), and of the southern slopes of the Alps), see for instance Huss, M., Dhulst, L., & Bauder, A. (2015). New long-term mass-balance series for the Swiss Alps. Journal of Glaciology, 61(227), 551-562; you will see that, at least for Switzerland, there are quite a few long-term mass balance time series that you could use… For Austrian measured mass balances, you could also contact the WGMS national correspondent Andrea Fischer, for Norway Liss Andreassen (NVE), for western Canada, I am sure Brian Menounos (University of Northern British Columbia) would be willing to help you out with further detailed information. Have also a closer look at the detailed database of measured mass balances worldwide provided through [www.wgms.ch](www.wgms.ch). As this is, in my opinion, an important part of your study, it would be worth spending some more time and effort here I think, always aiming at taking long-term measured mass balance data with a regional and climatic context regarding individual catchments you analyze. – And if you have more than one mass balance time series to compare with one individual catchment, take area-weighted values!

- *Another general comment of the reviewer relates to the clarification of section 4.2, especially the header and Figure 3. The reviewer is right that section 4.2 only presents WD events; the other events are presented in section 4.3. As the reviewer suggests, we will add this in the title of section 4.2. The catchments in Figure 3 were chosen so that they represent the streamflow responses for catchments with different glacier cover, ranging from low to high. The example years were selected because in those years a warm and dry event occurred in multiple catchments, and in a specific month. We wanted to show an example for an event in September (European Alps 1985 in this case), one in August (Canada in 1981 in this case) and one in July, but found that in 2006 in Norway, events in multiple months occurred, therefore nicely illustrating responses, trends and C values for different months and in a catchment with different glacier cover. Figure 3 is meant as an example to illustrate what we analyze in this study and to show the range of responses, regarding streamflow trends during the events, compensation levels, and dependence on glacier cover. We will add the explanation of this selection of example catchments and years to the revised manuscript.*

**Reply:** Ok, I see, very good if you add some explanation and information thereupon in the revised version of the manuscript.

- *The reviewer indicates that it is unclear how we calculated correlations between changes in C and glacier changes over time, referring to Figure 10. The phrasing here might have caused some confusion, and we will clarify that in the revised manuscript; we do not correlate C with glacier changes, but analyze trends of C over time. Maybe the relative glacier cover on the x-axis is confusing here. Time trends were calculated for each catchment, so glacier cover, in this case, refers to the trends for different catchments, not to trends in glacier area or glacier cover of the catchment.*

  *To calculate trends, correlations can be used. For each catchment and each month, we checked if there were 8 or more events occurring in the time series. If so, the correlation between C and the year of the event was calculated. A negative value indicates decreasing C over time, and a positive correlation means an increasing level*

*of compensation (C) over time. A decreasing trend may be attributed to retreating glaciers.*

*We agree with the reviewer that in some regions, glaciers were not only retreating but also had intermittent phases of advance (we looked at glacier length data from WGMS and literature). However, since only a few (extreme) events occur in the time series, limiting the time series only to phases of glacier retreat, will result in too few events to do a trend analysis. On page 22 L438-449 (discussion, 5.5), we will add a discussion on the phases of glacier advance and their possible effects on trends in C. These effects of phases of glacier advance on C may, however, not be straightforward. Also, an event can occur in this advancing phase, or not because these periods are generally characterized by colder temperatures, possibly influencing the effect of such phases on the overall trend.*

**Reply:** I am ok with how you want to implement this comment and thank you for your clarifications here.

- *The last general comment relates to the use of 'Norway' and 'Canada', while actually only specific regions within these countries are analyzed, namely southwestern Norway and western Canada. We agree with the reviewer and will change 'Norway' and Canada' into 'southwestern Norway' and 'western Canada' in the revised manuscript.*

**Reply:** Great, I think this is quite important, many thanks!

*Specific comments:*

*We thank the reviewer for all detailed comments and technical corrections which will greatly help to improve the paper and its clarity. We will consider them to revise the manuscript carefully.*

**Reply:** You're welcome, thank you! As I wrote above, I would very much appreciate if you could also provide a document with point-by-point comments on how you implemented all the specific and technical comments (also regarding figures and tables) of my review in the end to the public discussion (in order to be transparent and provide complete author's responses). Kind regards and all the best ☺ Dr. Mauro Fischer

[Figure]

*Figure 1 Comparision of the correlation between country-wide mass balances and level of compensation (C), left: using the median to average mass balance observations and create a country-wide mass balance time series (Figure 9 in the manuscript), right: using area-weighted average to calculate a country-wide mass balance time series. Results are shown for the European Alps.*

---

## Referee Comment (RC3)

**Addendum to reviewer reply on AC1-hess-2021-44 by Mauro Fischer, University of Bern**

*Regarding the use of these regional/country averages mass balance time series, we agree that this is not the best approach. Still, we are rather limited because of limited observations. The ideal option would be to have long-term mass balance time series available for all glaciers in each catchment. In reality we are far from that situation. The reviewer suggests to not average mass balance time series per country because of significant regional climatic conditions. Using geodetic mass balance data, or mass balance data calculated from a model, as done in the studies referred to by the reviewer, is not really an option here, as we need annual variations and would like to base this study solely on observations rather than mixing observations and simulations. We would thus need to find a way to extrapolate the few observations that exist, to all the other glaciers in the studied catchments. The study of Huss (2012) nicely illustrates different options to do that, i.e. arithmetic average, using glacier hypsometry or multiple regression. Although Huss did such an analysis for the European Alps, doing something similar for southwestern Norway and western Canada is beyond the scope of this study. The only option left would be to find out which catchments have a measured glacier in their boundaries that has long-enough time series to do a correlation analyses with the level of compensation. Area-weighted averages will not need to be used then, as it is unlikely that more than one glacier in the same catchment is measured. This will reduce the number of catchments analysed in this part of the study significantly, but may represent better the relation between mass balances and level of compensation and will be worth testing.*

**Reply:** I am totally aware of the fact that long-term measured mass balance data is lacking for a lot of catchments you analyzed. Still, at least for areas with comparatively (spatially) very dense and rather long-term measured mass balance time-series like Switzerland or Austria (or maybe even southwestern Norway?!), I think it would be worth taking only mass balance data from glaciers with "comparable regional climate conditions" for analyses of your catchments (e.g. differentiate between catchments of the northern slopes of the alps, of the inner (high) alpine regions (there also between west and east), and of the southern slopes of the Alps), see for instance Huss, M., Dhulst, L., & Bauder, A. (2015). New long-term mass-balance series for the Swiss Alps. Journal of Glaciology, 61(227), 551-562; you will see that, at least for Switzerland, there are quite a few long-term mass balance time series that you could use… For Austrian measured mass balances, you could also contact the WGMS national correspondent Andrea Fischer, for Norway Liss Andreassen (NVE), for western Canada, I am sure Brian Menounos (University of Northern British Columbia) would be willing to help you out with further detailed information. Have also a closer look at the detailed database of measured mass balances worldwide provided through www.wgms.ch. As this is, in my opinion, an important part of your study, it would be worth spending some more time and effort here I think, always aiming at taking long-term measured mass balance data with a regional and climatic context regarding individual catchments you analyze. – And if you have more than one mass balance time series to compare with one individual catchment, take area-weighted values!

**Addendum by reviewer 22.02.2021**
Dear Marit and Co-authors, what I wanted to add here is that of course you will have to consider the glacier size class distribution of the catchments you analyze in order to choose which and how many long-term mass balance time series you take into account for your new calculations. Example: If you have mass balance data for a small glacier in one catchment you analyze, but the glacier size class distribution of the catchment is more "towards larger glaciers", i.e. there are also larger glaciers in the catchment, then it would be wrong as well to only take mass balance data from this single small glacier situated in the catchment you analyze (as catchment-wide mass balances will be strongly influenced by larger glaciers)… So to add on my comment above here: I would just try to take as many long-term mass balance data as you have for glaciers in the same region and with more or less the same climatic conditions as for the catchments you analyze, and then take area-weighted values of these to compare them with *C* values of the catchment you analyze (or you use other proposed approaches to extrapolate measured mass balance data to specific regions or catchments, as for instance discussed in Huss, M. (2012). Extrapolating glacier mass balance to the mountain-range scale: the European Alps 1900–2100. The Cryosphere, 6(4), 713-727.). Kind regards and all the best, Mauro

---

## Author Comment (AC2)

HESS-2021-44

Reply to Anonymous reviewer #2

**Review of "Hydrological response to warm and dry weather: do glaciers compensate?" submitted by Marit van Tiel, Anne F. Van Loon, Jan Seibert, and Kerstin Stahl to Hydrology and Earth System Sciences (HESS)**

I found this study interesting and well-written. The findings are novel and the methodology is clear. My one concern is with the portion of the study dealing with glacier mass balance – which has already been brought up by in the review by Mauro Fischer, see my further general comment on this topic below.

The detailed review by Mauro Fischer caught most of the minor and technical comments that I would have included in my review. To save the authors time in responding to duplicate comments, I will not repeat them here. The few comments below are those I have tried to prune for overlap with Mauro's review. I enjoyed reading this study and, in my opinion, it is worthy of publication in HESS with some minor revision.

>> *We thank the reviewer for this positive evaluation. Please find below our replies in blue and italic.*

**General Comments**

Like the other reviewer, I have concerns about using the median of measured mass balance data. Using the area-weighted average, as has already been suggested, is a better option and has already been demonstrated in the response from the authors. I recognize the data limitations the authors are contending with, and I wonder if the mass balance analysis could just be removed from the study. I understand why the authors would like to include this type of analysis, but, in my opinion, it is a very minor part of the study and novelty and importance of the manuscript would not be hindered by removing this small piece. It would be a great avenue for future research. I will leave it up to the authors to decide whether a revised version of the mass balance analysis with the area-weighted averages should or should not be included in the revised manuscript.

>> *Yes, both reviews made us aware that area-weighted mass balances should be used instead of the median. Given the data limitations, the reviewer suggests leaving the analysis out of the current study. We agree with the reviewer that it is only a minor part of the study. The study illustrated that on these spatial and temporal scales, the available glacier mass balance data has varying relations with the compensation level and attribution is difficult. We agree that this attribution and the connection between glacier mass balance data and streamflow responses is an excellent avenue for future research.*

>> *Now we have two opinions: we can follow reviewer #1, indicating that it is an important part of our study and we should use area-weighted mass balances time series, preferably subdividing the glacier mass balance observations and catchments into more climatically similar regions (northern and southern slopes of the Alps, west and east). Alternatively, we can follow reviewer #2 who suggested leaving this part of the study out because it is only a very minor part.*

>> *We have a slight preference for leaving the glacier mass balance analysis out of the study and will change the manuscript accordingly.*

**Specific Comments**

Discussion:

Two sections in the discussion seem to overlap: '5.3 Drivers of Event-to-event variability in compensation levels', and '5.5 Temporal variability in event responses'. Both sections are discussing results presented section 4.4 (drivers of event-to-event variability). From the headings, it is not clear to me what the difference is between the two sections. I suggest combining the two sections into one.

*>> Indeed, both sections discuss the findings of Section 4.4. In 5.3 we discuss the effect of the conditions of the event on the compensation level (antecedent conditions, duration and temperature). In 5.5 we discuss temporal effects, thus the difference between the different months and the impact of trends. However, since the conditions also vary for the different months, the difference between the two subsections may have become obscure. We suggest to make one section out of it called 'Drivers of event-to-event variability in compensation levels and monthly differences', in which we combine the conditions and the monthly/seasonal effects and add the trend discussion.*

There are several places in the discussion where I found myself wanting a reference back to the relevant results. Below I've listed two locations. In my opinion, cohesion through the manuscript would be improved by adding a few figure, table, or section references to the discussion.

Line 381 – refer to the results supporting this stated finding.

Line 438-439 – refer to the results supporting the stated finding

*>> Thank you for pointing that out. In the revised version, we will add references to figures, tables and preceeding sections to the discussion.*

**Other minor comments:**

Figure 1 and Figure S1:  Are there just two line thicknesses used in these figures? What gc values do these thicknesses correspond to? What is the break value?

*>> Do you mean Figure 2 and Figure S1? The link thickness scales with the relative glacier cover, so there are many different line thicknesses. We will adjust this in the legend and caption to make it clear.*

Figure 2 – Add a legend to clarify the relation between color and region.

*>> We will add this.*

Figure 8 – What are the glacier cover classes? Perhaps just list them in the caption.

*>> The glacier cover classes are given in the top-left figure. We will add them in the complete first row or, indeed, list them in the caption to make it more clear.*

Table 3 – Heading should be 'Average duration of events [d]'?

*>> Indeed, this will be changed in the revised version*

Line 270: Missing comma in sentence starting with 'All variables,'

Line 393: 'and follow up studies Moyer et al. (2016)).' Seems like part of this sentence is missing

*>> Thank you for pointing these things out. We will change this in the revised version*

---

## Author Comment (AC3)

**Reply to RC2 and RC3 hess-2021-44**

**RC2**: I am totally aware of the fact that long-term measured mass balance data is lacking for a lot of catchments you analyzed. Still, at least for areas with comparatively (spatially) very dense and rather long-term measured mass balance time-series like Switzerland or Austria(or maybe even southwestern Norway?!), I think it would be worth taking only mass balance data from glaciers with "comparable regional climate conditions" for analyses of your catchments (e.g. differentiate between catchments of the northern slopes of the alps, of the inner (high) alpine regions (there also between west and east), and of the southern slopes of the Alps), see for instance Huss, M., Dhulst, L., & Bauder, A. (2015). New long-term mass-balance series for the Swiss Alps. Journal of Glaciology, 61(227), 551-562; you will see that, at least for Switzerland, there are quite a few long-term mass balance time series that you could use... For Austrian measured mass balances, you could also contact the WGMS national correspondent Andrea Fischer, for Norway Liss Andreassen (NVE), for western Canada, I am sure Brian Menounos (University of Northern British Columbia) would be willing to help you out with further detailed information. Have also a closer look at the detailed database of measured mass balances worldwide provided through www.wgms.ch. As this is, in my opinion, an important part of your study, it would be worth spending some more time and effort here I think, always aiming at taking long-term measured mass balance data with a regional and climatic context regarding individual catchments you analyze. –And if you have more than one mass balance time series to compare with one individual catchment, take area-weighted values!

==RC3 Addendum by reviewer 22.02.2021==

Dear Marit and Co-authors, what I wanted to add here is that of course you will have to consider the glacier size class distribution of the catchments you analyze in order to choose which and how many long-term mass balance time series you take into account for your new calculations. Example: If you have mass balance data for a small glacier in one catchment you analyze, but the glacier size class distribution of the catchment is more "towards larger glaciers", i.e. there are also larger glaciers in the catchment, then it would be wrong as well to only take mass balance data from this single small glacier situated in the catchment you analyze(as catchment-wide mass balances will be strongly influenced by larger glaciers)... So to add on my comment above here: I would just try to take as many long-term mass balance data as you have for glaciers in the same region and with more or less the same climatic conditions as for the catchments you analyze, and then take area-weighted values of these to compare them with C values of the catchment you analyze(or you use other proposed approaches to extrapolate measured mass balance data to specific regions or catchments, as for instance discussed in Huss, M. (2012). Extrapolating glacier mass balance to the mountain-range scale: the European Alps 1900–2100. The Cryosphere, 6(4), 713-727.). Kind regards and all the best, Mauro

>> *We thank the reviewer for his clarification and suggestions regarding the mass balance analyses. In fact, we did use the WGMS database, all the observations that are in there were used.*

*Considering review 1 and review 2, we now have two options for the glacier mass balance analysis: we can follow reviewer #1, indicating that it is an important part of our study and we should use area-weighted mass balances time series, preferably subdividing the glacier mass balance observations and catchments into more climatically similar regions (e.g. northern and southern slopes of the Alps, west and east). Alternatively, we can follow reviewer #2 who suggested leaving this part of the study out because it is only a very minor part. We have a slight preference for leaving the glacier mass balance analysis out of the*

*study and plan to change the manuscript accordingly. We will await the editor's guidance to this and make a final decision then.*

---

## Author Response (AR1)

Dear Editor,

We would like to thank the two reviewers and the editor for the feedback on our manuscript. Please find our detailed answers to the reviewers' comments and the changes we have made below (responses in italic blue). A track change version of the manuscript is also included. We excluded the mass balance analysis from the manuscript and clarified the text, as suggested by the reviewers. We merged two sections of the discussion. Page and line numbers refer to the track changed version of the manuscript.

##############################################################################

Review #1

**General comments**
I want to thank the authors for their noticeably large amount of work for this very nice study and valuable scientific contribution. In my opinion, the study contains novel and highly interesting findings, the presented results seem solid overall (except maybe for how the authors calculated catchment-wide mass balance, see comments below) and are of high interest to the scientific community. In general, the manuscript is well written and very nicely illustrated (great figures!). I think that this work clearly deserves to get published in Hydrology and Earth System Sciences. The authors will need to put some effort into correction and improvement of their manuscript. In my opinion, there are still (even if only few) important major and a number of minor issues which need to be addressed, corrected, clarified, and implemented prior to publication. I guess that the majority of the specific comments listed below are easy to implement, whereas some general or specific comments will maybe need some additional work and time. I hope that my work for this review will help improving the paper, and I encourage the authors to implement and reply to all my comments as far as possible. Thanks a lot, it was a pleasure to review your study, congrats, all the best and kind regards.

**List of some general comments:**
-Abstract: very well written and very good overall! See some specific comments below. As you analyzed glacier-fed streamflow responses to WD in long-term streamflow observations (>50 years), in my opinion it would be beneficial for people interested in your paper to add some information about your observation periods for Canada, Norway, and the European Alps directly in the abstract as well...

>> *We agree that this is a good idea. However, the different catchments have all different observation periods. For the Alps, streamflow and meteorological data overlap for the period 1961-2015/2016. Still, for Norway and Canada, individual catchments have a different observation period (but all at least 50 years), and therefore it is not easy to summarize that in the abstract.*

-Introduction: I like the structure/storyline and content of the introduction, maybe it is a bit (too) long and could be shortened without losing important information and content? Sometimes I wondered whether the difference between "the buffering effect of glaciers" and "the compensation effect of glaciers" is always clear (or if there is any big difference at all...). Maybe you could check that point while going through the introduction (or the whole manuscript) again? Moreover, please find a number of specific comments which hopefully help to improve specific parts of the introduction below..

>> *In our opinion, these concepts mean something different but are in the literature often used interchangeably when describing glaciers and droughts. We explain this in the discussion and added the following text in the introduction:*

*To distinguish these different buffer characteristics, we refer to compensation processes when describing the active role of glacier melt adding additional water and buffering when*

*describing the general function of catchment storages providing water when precipitation input is low. Such a distinction is important to quantify the effect of glacier melt during dry periods (L51-54).*

*In our view, buffering means 'something that helps protect from harm', i.e., it is usually a passive process but could also be an active process. In our context, glaciers act as a buffer to (meteorological) droughts because they still provide water while rainfall input is low, thereby preventing a severe impact on water availability, especially in relative terms. In this view, glaciers are always a buffer, regardless of the situation, because their water supply (on the short-term) is dependent on temperature and radiation rather than rainfall. Snow and groundwater can therefore be seen as buffers to meteorological droughts as well. 'To compensate', in our view, means to provide something to reduce the effect of something that has been lost or damaged, i.e., it is an active process. In our context, it means that excess (more than normal) glacier melt can reduce the effect of a rainfall deficit on streamflow. In the case of buffering, the rainfall deficit in streamflow can still be present, but streamflow will not be affected entirely, because part of the streamflow comes from groundwater/snowmelt/glacier melt. In case of compensation, excess glacier melt can compensate for the rainfall deficit, and streamflow can be close to average or above average levels. Since glaciers do not only buffer but also compensate during warm and dry events, we argue (in the discussion) that such a distinction should be made and that this approach helps to quantify this specific role of glaciers.*

*To shorten the introduction, we removed one of the paragraphs (L90 – 102)*

-Data: I have, unfortunately, some concerns about how you deal with and use measured mass balance data in your study; you cannot just take a median of measured glacier mass balance data, you have to weight the measured data with glacier area, i.e. calculate area-weighted average mass balances, otherwise you might get wrong results. This is relevant when it comes to compare or correlate mass balance with levels of compensation C, I clearly think you need to check and clarify that; moreover, it does not really make sense to me to just use "country-wide" mass balance data for individual catchments you analyze, there are better ways to extrapolate measured mass balance data to individual catchments, especially in areas with a high density of measured mass balance data like Norway, Switzerland, Austria. Please see also specific comments thereupon below.

*>> This part of the analyses is now removed from the manuscript*

-Results: In section 4.2 you only refer to WD (or WWD) events, right (CD, and ND events are excluded)? Maybe make that clearer in the text (e.g. by changing the title of the section to "4.2 Glacier compensation during WD events in different catchments"); it was not very much clear to me why the chosen catchments and years for individual regions were selected for Figure 3 (just an example or do these three years and catchments reflect some of the "general observations, trends, and C values over the whole observation period"?)...

*>> The reviewer is correct that section 4.2 only presents WD events; the other events are presented in section 4.3. We have added this in the title of section 4.2. The catchments in Figure 3 were chosen to represent the streamflow responses for catchments with different glacier cover, ranging from low to high. The example years were selected because a warm and dry event occurred in multiple catchments in those years, and in a specific month. We wanted to show an example for an event in September (European Alps 1985 in this case), one in August (Canada in 1981 in this case) and one in July. Still, we found that in 2006 in Norway, events in multiple months occurred, therefore nicely illustrating responses, trends and C values for different months and in a catchment with a different glacier cover. Figure 3 is meant to illustrate what we analyze in this study and show the range of responses regarding streamflow trends during the events, compensation levels, and dependence on*

*glacier cover. This explanation of this selection of example catchments and years was added to the revised manuscript (L280).*

-Results: Section 4.4and figure 9: As I understood how you calculate catchment-wide mass balance or take into account measured mass balance data to correlate it with levels of compensation C, in my opinion this is not quite correct (see comments above and specific comments thereupon below), maybe by correctly calculate catchment-wide mass balance your resulting correlations of mass balance with levels of compensation would quite change and show another relation between these variables?!...

*>> This part of the analyses is now removed from the manuscript*

-Results: Still section 4.4 and figure 10: To be honest, I didn't really understand how you calculated correlations between changes in C and glacier changes over time (mostly retreat for your period of observation, but with intermittent phases of readvance, for instance for Norwegian glaciers in the 1990s!), this is neither very clear from the methods section nor very clear in the results section... what data sets did you take into account to check if glaciers in the analyzed catchments did change in area and volume over your observation period? – this aspect, relating also to figure 10, is not very clear...

*>> The phrasing here might have caused some confusion, and we have clarified that in the revised manuscript; we do not correlate C with glacier changes, but analyze trends of C over time. Maybe the relative glacier cover on the x-axis is confusing here. Time trends were calculated for each catchment, so glacier cover, in this case, refers to the trends for different catchments, not to trends in glacier area or glacier cover of the catchment.*

*To calculate trends, correlations can be used. For each catchment and each month, we checked if there were eight or more events occurring in the time series. If so, the correlation between C and the year of the event was calculated. A negative value indicates decreasing C over time, and a positive correlation means an increasing level of compensation (C) over time. A decreasing trend may be attributed to retreating glaciers. We agree with the reviewer that in some regions, glaciers were not only retreating but also had intermittent phases of advance (we looked at glacier length data from WGMS and literature). However, since only a few (extreme) events occur in the time series, limiting the time series only to phases of glacier retreat will result in too few events to do a trend analysis. In the Discussion, Line 457, we added a sentence on the possible influence of different phases of glacier retreat and advance on time trends in C.*

-In the whole manuscript: I think it would make more sense to use "southwestern Norway" instead of "Norway" and "western Canada" instead of "Canada" in the entire manuscript (these countries are so large and you analyze only catchments in specific regions)...

*>> we have changed 'Norway' and 'Canada' into 'southwestern Norway' and 'western Canada'*

**Specific comments and technical corrections**

To facilitate the author's correction of the manuscript I combined specific comments and technical corrections (including language or spelling and comprehensibility issues). Sometimes comments contain both specific comments and technical corrections, sometimes just one or the other. I would ask the authors to implement my comments and suggestions as far as possible.

*>> Thank you for the detailed specific comments and technical corrections. We changed all of them. Please find below all reviewer comments and our response where there was more than a simple change..*

**Title**

Personally, I am also for short and concise, attractive titles (and I like yours), but maybe it would be good to add short information about your period of observation and study areas in the title? Up to you...

*>> We decided to keep the title, to not make it too long. The study areas and observation period are in the abstract.*

**Abstract**

Lns11f: "C was, in general, higher than 100% for catchments with a higher relative glacier cover" is it possible to give a number of the (average) relative glacier cover (XY% of the catchment glacierized over the observation period) needed to result in a C≥100%? Would be quite interesting...

*>> We added that information from the conclusion to the abstract.*

Lns17ff: I know that you have to summarize and make short statements in the abstract, but, in my opinion, the last two sentences of the abstract could be a bit clearer and more precise (how do glaciers not compensate straightforwardly? What are "the different streamflow contributions and their variations" you refer to here?)

*>> changed into: Overall, our results suggest that glaciers do not compensate straightforwardly, and the range in compensation levels is large. The different streamflow components, glacier, snow and rain, and their variations are important for the buffering capacity and the compensating effect of glaciers in these high mountain water systems.*

**Introduction**
Ln 31: "..., because of temperature-driven water supply." why not directly write what you refer to here? i.e. enhanced snow and ice melt due to high temperatures?...

*>> Indeed, we refer here to enhanced snow and ice melt. However, we want to indicate the contrast here between precipitation-driven water supply and temperature-driven water supply. Regions that also have temperature-driven water supply can be less prone to meteorological droughts (lack of precipitation). For clarification, we added '…., instead of only precipitation-driven, water supply (L34).*

Lns42f: "Hence, groundwater and snow storages might not always be a perfect buffer during warm and dry periods." Please be a bit clearer and more precise here, I guess what you want to say here is something like "Hence, groundwater only has a limited buffering capacity (in terms of runoff, provides only baseflow), while the buffering capacity of snow is higher (in terms of additional runoff) but temporally limited (if all snow has melted, there is no buffer anymore).
*>> changed*

Ln 55: "...from other streamflow contributions such as snowmelt." maybe add some additional important streamflow contributing processes (in high mountain areas) here "...from other streamflow contributions such as snowmelt, surface runoff, interflow, groundwater flow or melting permafrost."?!?!
*>> changed into 'such as snowmelt, rainfall-runoff and groundwater'*

Lns 60ff: "During warm and dry years, glaciers can provide more meltwater to streamflow, and during cold and wet years they generate less meltwater so that altogether the interannual streamflow variability is relatively low." do the studies you refer to here give a "lower threshold ratio of glacierization" for individual catchments (i.e. minimum percentage of glacier-covered area in a catchment) from which the dampening effect of glaciers on interannual streamflow variability applies? –would be interesting added value here...

*>> The studies referred to here show that there is a convex relationship between streamflow variability and glacier cover, i.e., there is no lower threshold but rather an optimum. However, van Tiel et al. (2020a) discuss that such an optimum is not clearly distinguishable looking at observed data. To give a rough idea, we added here that the optimum is usually found between 10 and 40% relative glacier cover in different studies.*

Lns66f: "...but also other climate and catchment characteristics appear to influence the streamflow sensitivity to climatic anomalies..." other climate and catchment characteristics like what? –can you give some examples mentioned in the studies you refer to here?
*>> added*

Ln 72: "...provided an additional source of water in the summer compared to the seasonally delayed contribution." why "COMPARED TO the seasonally delayed contribution"? rephrase in order to be clearer...
*>> changed into: in addition to*

Lns80ff: "However, this contribution may reach a maximum, as the higher glacierized catchments are generally located at higher elevations that receive more orographic precipitation amounts..." can you please check that statement? In my opinion, it is not primarily the increased amounts of precipitation that causes a maximum value of relative catchment glacier cover in terms of increased streamflow contribution by glacier melt, but rather air temperature and climatic conditions in highly glacierized catchments, isn't it?

*>> We are not entirely sure what the reviewer means here. Higher glacierized catchments can be at higher elevations, but also the gauging station can be located closer to the glacier. In the latter case, the climatic conditions on the glacier do not change. Overall, these may affect the total glacier melt contributions, but less in a relative way. Anyway, we removed this part of the introduction to shorten it.*

Ln 83: "...is generally assumed to be highest in August and September, ..." of course, this is only true for mountain glaciers in the northern hemisphere with a mass balance regime similar to the one in the Alps... I am sure you are aware of that but it might be good to be more precise here... (as your study is interesting to people from all over the world ;-))

*>> indeed, but this paragraph is now removed from the introduction.*

Lns99f: "...we analyzed observed hydrological responses to WD events for catchments with varying glacier cover in Norway, Canada, Switzerland and Austria." I recommend to add some information about the observation periods / temporal time frame of your study here

*>> we added the overall period. In the table in the supplementary material, the individual periods for each catchments can now be found.*

**Data and hydroclimatology of selected glacierized catchments**
Lns 109f: "A few of these catchments are nested." What does this exactly mean here? –Can you maybe briefly explain in key words in parentheses after the sentence?

*>> We have added:*
*(meaning part of the meteorological conditions and streamflow responses are similar)*

Ln 110: "...derived from gridded data products..." "...derived from gridded reanalysis data..."?
*>> No, we used here gridded data that comes from the interpolation of meteorological observations – we added this information in parentheses*

Ln 118: "...but the selection of events..." "...but the selection of WD events..."?
*>> We changed into dry events because all types of events could only be selected until 2012 in western Canada.*

Lns 122f: "Therefore, regional average mass balance time series were calculated from all available mass balance observations per year per country (Austria, Switzerland, Norway and Canada) by taking the median." If I understood your approach right, then, as a glaciologist, I have some concerns here: 1) Taking just an average or median value from measured mass balance data of individual glaciers is not really correct, you need to calculate area-weighted average mass balance values (as for instance large valley glaciers have their ablation area at lower elevation (therefore more melt, therefore more negative mass balance) compared to smaller mountain glaciers having their terminus at higher elevation (therefore less melt, therefore less negative mass balance)) ;i.e. you have to multiply all mass balance data you take into account by glacier area, then calculate the sum of these values for all glaciers, and finally divide that sum by the summed up glacier areas 2) Even though for instance Switzerland and Austria are small countries, measured in-situ mass balance sometimes comes from glaciers with significantly different regional climatic conditions (the same must apply for Canada, for Norway your catchment sample lies in a more or less similar climatic zone I guess)... So my point here is that, in my opinion, it does not make sense to calculate average mass balance values "by country" for the catchments you analyze! For example, Schaefli et al. (2019, The role of glacier retreat for Swiss hydropower production, Renewable Energy) calculated area-weighted mass balance data for all glacier-covered catchments with hydropower plants using data by Fischer et al. (2015, Surface elevation and mass changes for all Swiss glaciers 1980-2010, The Cryosphere). I think you should find a way to reasonably derive catchment-wide mean annual mass balance data from existing measured mass balance data and extrapolation techniques(for the latter see for instance Huss (2012), Extrapolating glacier mass balance to the mountain-range scale: the European Alps 1900–2100,The Cryosphere); you might also work with Huss and Hock (2015, A new model for global glacier change and sea-level rise, Frontiers in Earth Science, there, past and future glacier mass balance data is available(per glacier!)from the Global Glacier Evolution Model (GloGEM)). Whatever data you use or method you apply to extrapolate (measured)mass balance data to glaciers of your analyzed catchments, do not forget that you have to calculate area-weighted values again if you have to work with "catchment-wide" mass balance data for your analyses. If you want you can also contact me personally (mauro.fischer@giub.unibe.ch) and I would be happy to discuss that with you and try to help you there.

*>> The glacier mass balance analyses were removed from the analyses.*

Ln 128: "...(GI4, 2015)..." I think you need to add the correct reference for the Austrian glacier inventory used here.

*>> we added the correct reference*

Ln 128: "...ranging in size from..." rather put the "in size" at the end of the sentence (after the numbers)

*>> Ln 141? changed*

Ln 174: I would delete the "(not shown)" here... but would it maybe make sense to add a table with catchment name, location, size, mean elevation, mean annual temperature, mean annual precipitation and resulting levels of compensation to the Supplementary Materials? – This would add some relevant detail for people interested in catchments you looked at..

*>> We added a table in the supplementary material, showing the catchments and their characteristics.*

**Methods**

Lns 202ff: "It was assumed that in these high elevation catchments there is an immediate response to the WD events (or the other dry events) and streamflow data of the dates exactly corresponding to the events were selected." can you add some information on why (from a process understanding point of view) it is ok to work with that assumption? I argue that you're right with this assumption but giving some rationale here would be good I think.

*>> Because we work with 7-day smoothed values of P, T and Q, any delayed effect of P on Q is averaged out. Still, some effects may be there, but as we discuss in the discussion, the delay is difficult to assess and may vary from catchment to catchment and from event to event. We changed this part now into (L222-224):*

*Once the WD (or other dry events) periods were selected based on the P and T data, the corresponding streamflow data were analyzed. For the sake of simplicity, we selected the streamflow event to start and end on the same dates as the meteorologically dry events, aware that this assumption is only an approximation. Due to the use of 7-day smoothing for streamflow and precipitation, precipitation events before the dry events may only have a limited effect on the 7-day mean streamflow signal during the dry events.*

Lns 227f: "...the importance of different event drivers." as for example?

*>> added examples: (e.g., antecedent streamflow conditions and winter precipitation)*

Ln 231: "...at least 8 or more events..." what is your rationale behind this threshold?

*>> To be honest, there is no clear rationale. If the threshold is much higher, then no relationships can be analyzed because there are few events. If lower, then n might be too low (is already relatively low/too low) for any statistical analyses.*

**Results**

Lns240f: "...and different numbers of catchments per region." I would also argue that they are not comparable due to different (hydro-)climatological settings of the individual catchments!?

*>> yes, but this information cannot be extracted because of the differences in a number of catchments and length of the time series.*

Lns 295f: "Different rainfall amounts..."; "...high rain amounts in summer..."; "...low rain amounts in summer..." I guess you refer to the climatological statistics here (how much rain

falls on average in one region in summer, cf. figure 2)... I would write that somehow in this sentence in order to be clear...
*>> we added: Different average summer rainfall amounts*

Lns 300f: "Also, the relative glacier covers of the Canadian and Alps catchments are more complementary than comparable in this sample of catchments." I believe to see your point here, but doesn't make this statement figure 7 and your rationale/descriptions here about the influence of the amount of average summer rainfall on the levels of compensation a bit obsolete?...
*>> we agree and therefore decided to add it in the discussion instead of in the results section. We still want to show the analysis because it is often assumed that dry and wet summer catchments have different responses because of a different relative glacier melt importance. However, it becomes clear that relative glacier cover is important as well so that a 1:1 comparison between Canadian and European catchments cannot easily confirm this hypothesis on the event scale.*

Lns 323f: "...when $MB_{sum}$ is larger (more negative)." "...when $MB_{sum}$ is more negative." as I wrote above, try to avoid speaking of larger/higher and smaller/lower mass balance, this is always confusing (use "more negative" and "more positive")...
*>> Thank you for the suggestion. This is now removed from the manuscript.*

Lns 328f: "Most significant trends were found in June (Canada) and September (European Alps), which were all negative. Norwegian catchments showed mostly positive trends, except in September." ok, what does that mean? Can you maybe relate these correlations to observed glacier changes in the analyzed catchments? For Switzerland, see for instance M. Fischer et al. 2014 (The new Swiss glacier Inventory SGI2010, Arcitc, Antarctic and Alpine Research, section Study Region), for Austria, see for instance A. Fischer et al. 2015 (Tracing glacier changes in Austria from the Little Ice Age to the present using a lidar-based high-resolution glacier inventory in Austria, The Cryosphere), for Norway see for instance Winsvold et al. 2014 (Glacier area and length changes in Norway from repeat inventories, The Cryosphere), for western Canada see for instance Bolch et al. 2010 (Landsat-based inventory of glaciers in western Canada, 1985-2005, Remote sensing of Environment)..

*>> The reasons for trends in C were already included in the discussion, but we extended it to discussing more explicitly the effect of glacier changes. See also comment before*

**Discussion**
Ln 346: "...as in these rapidly changing systems..." "...as it concerns rapidly changing systems..."?
*>> changed to: The latter might present some additional challenge because in these rapidly changing systems, the daily regime benchmark can change significantly over long periods*

Ln 351: "...in the summer shoulder season..." "...in midsummer..."?
*>> no, we mean not high summer (July and August) but the shoulder seasons, June and September*

Lns 366f: "But Figure 4 shows that there are exceptions." I guess you refer to catchments and C values in Canada here? –Thus, exceptions concern catchments in drier average summer climates (cf. Figure 2) with already low relative glacier cover? be more concrete/precise here...
*>> no, we mean here that Figure 4 shows that all the WD events show quite a large spread in compensation levels. On average, if glacier cover reduces to ~5% there is no full compensation anymore. But still, in some events, it can still be the case (gray bars in figure 4). We changed the sentence:*

*However, there are exceptions, for some catchments in Canada with a low relative glacier cover and for some events in all of the low glacierized catchments because the ranges in compensation levels can be large (Figure \ref{fig:04}), i.e. some events in low glacierized catchments can still result in $C$ levels above 100\% (L402-404)*

Ln 372: "...also make it difficult to answer when glaciers compensate because it depends on the situation." a bit vague and not very clear to me, can you rephrase in order to be more precise here?
*>> Changed into: These wide ranges hamper a clear conclusion on the question in which situation do glaciers compensate because it does not only depend on catchment characteristics but also on the specific situation in which the event takes place (L408)*

Ln 376: Why do you refer to Figure 11 here? –Shouldn't you refer to figure 8?
*>> We refer to Figure 11 here to explain the concept of different streamflow contributors. We now also refer to Figure 8.*

Ln 384: "...became less clear..." "...is less clear..."?
*>> Over the summer, the relation in their study became less clear. Became refers to a changing process here over the summer.*

Ln 392: Can you shortly explain in parentheses what "carry-over storage" exactly means?
*>> We added this in parentheses: (storage from the previous period that is released in the current period)*
*For example, if streamflow is high in July – it can still have an effect on streamflow in August because of water stored in a 'wetter' July, slowly releasing into August. Or high July streamflow may relate to extreme snow accumulation years, resulting in more streamflow in July and August (Moore et al., 2020)………*

Ln 400: Why do you refer to Figure 11 here? –Shouldn't you refer to figure 8?
*>> See other explanation*

Lns418f: "Also, increasing high temperatures or more often occurring relatively extreme temperatures will not be sustained with higher melt contributions when glaciers retreat." this is only true for time periods after "peak water" of individual catchments, I am sure you know that, but maybe you could include that in some way in your statement here...
*>> Yes, with not sustained we mean in the long run. We added: 'when glaciers have retreated considerably'.*

*Peak water, often referring to annual or summer streamflow or glacier runoff changes over time, if it exists so clearly, may not be directly transferable to responses to extremely warm and dry events. There may still be higher melt contributions during such extreme events, despite an overall post-peak streamflow trend. But we do know, in the long run, retreating glaciers will also reduce such event glacier melt contributions. But how to relate extreme events and the overall peak water concept is not yet clear, and therefore we would argue not to include peak water in this statement here.*

Ln 421: "...400 mm higher." I am not sure how I can interpret this value, is it per $m_2$(for the entire catchment)? Add some information here...
*>> 400 mm are for the catchment, yes. Per unit area of the catchment, streamflow was 400 mm higher when the glacier outline of 1979 was used. We added that this value refers to the catchment scale.*

Ln 513: "...there is quite a risk of internal model compensation..." can you add some information here (maybe in parentheses)? What does that actually mean? What exactly do you refer to here?

*>> We added: (e.g., lack of precipitation or snowmelt, due to input or model process uncertainties could be compensated for by extra glacier melt)*

**Figures**
Figure 1: Maybe add scales to a), b), c). I would write "Norway" (in green), "European Alps" (in blue), and "Western Canada" (in pink) at the lower right of the figure 1d). Moreover, can you add a more detailed legend for the circle sizes which signify the relative glacier cover of a catchment (i.e. how much relative glacier cover do the different circle sizes in 1d) represent?). Moreover, would it possibly make sense to additionally draw areas with "similar hydroclimatological regimes" (cf. chapter 2.3) in Fig. 1? –Maybe this would add valuable graphical information...

*>> We made a new version of the figure, including scales and adding the three regions in the lower right of figure 1d. We added an explanation about the circle sizes in the caption. We added the hydroclimatological regimes in the overview table of all catchments, see the new SI table.*

Figure 2: Looking at figure 2 I cannot really understand why two different graphs for Norway, four different graphs for Canada, and three different graphs for the European Alps. Can you please add something on that in the figure caption? You explain it in chapter 2.3 but I think to add some information thereupon in key words would be good for the figure caption.

*>> we added: The precipitation distribution over the year of the different catchments were grouped into several precipitation regimes per region: two in southwestern Norway, four in western Canada and three in the European Alps.*

Figure 3: You have to add a bit of information in the figure caption there: Write what exactly the i) black line, ii) blue line, and iii) the red dots signify (I know it's written in abbreviations at the bottom of the graph but it would be helpful to have that information (as text) in the figure caption as well). The same for the relative glacier cover (bold black number in the upper right corners), write it as text in the figure caption as well to be clear. Can you please also add somewhere in the manuscript (text or figures) based on what you chose to show the selected results for some selected catchments and selected individual years for the three regions (shown in Figure 3)? What was your rationale behind that? And maybe also which catchments (names, location) are shown?

*>> Changed into: Examples of streamflow responses ($Q_{r}$ in red) to WD events in different catchments and in different months. Each row represents one region and one specific year. The columns are different catchments, which are sorted from low to high relative glacier cover (left to right). The relative glacier cover of the catchments is indicated in the top-right corner. The black line shows the 7-day smoothed streamflow in the summer of the respective year and the blue line shows the long-term daily regime. Note the different y-axis for some of the plots.*

Figure 4: Figure caption: Write "Mean catchment level of compensation..." as in the figure caption of Figure 6!?;"...
*>> No, this would not fit here. In Figure 4 we show the mean catchment level of compensation (in colors), but also the ranges. In Figure 6 we only show the mean level of compensation for each catchment to all events.*

Figure 8: Figure caption: "Explained variance of C..." "Explained variance in C..."; moreover, neither from the figure nor from the text it is evident how the thresholds to separate individual classes of relative glacier cover were chosen... can you add some information on that please?

*>> We added in the methods section: The grouping was done based on the glacier cover distribution of the catchments and previous findings that around 10\% glacier cover streamflow sensitivities to precipitation and temperature variations change \cite{VanTiel2020}*

Figure 9: Figure caption: Delete the "9" at the beginning of the figure caption; you would have to write out the used abbreviations for annual, winter and summer mass balance somewhere (I would argue better in the text than here in the figure caption, for $MB_{win}$ you did it in the text);"Colours and symbols indicate the three regions (as in other figures)." why not be concrete and directly write colours of symbols together with corresponding regions here; can you explain the "$r_s$" (y-axes)? I would just add ($r_s$) in the figure caption in parentheses after "Spearman rank correlation"...

*>> Figure is deleted from the revised manuscript*

Figure 10:Figure caption: I would again write colours of symbols together with corresponding regions here; moreover, it is not really clear to me how time trends of C were related to time trends of relative glacier cover here (see general comments thereupon above...), I would need some more information in the figure caption to know how to exactly interpret this figure...

*>> Changed into: Time trends of $C$ for each catchment calculated as Spearman Rank Correlation Coefficients ($r_s$) between $C$ and year of the WD event. Norwegian catchments are indicated in green, catchments in the European Alps in blue and Canadian catchments in pink. Circles indicate significant trends (a=0.05).*

**Tables**
Table 2: Table caption: "...level of compensation. (C)" "...level of compensation (C)."; "Higher elevated glaciers:" "Glaciers at higher elevation:..."; "Percentile of mean temperature in spring (MAM)" which percentile?; "Percentile of sum of precipitation in winter (DJF)" which percentile?; "The streamflow percentile of the 30 days before the WD event" which percentile?;

*>> There is no which in this case. To compare the different catchments and the different months, we did not look at absolute streamflow amounts or winter precipitation amounts but instead looked at percentiles. For each of these variables, the variable for that year was expressed as a percentile compared to the rest of the time series. Percentiles are just used here as an anomaly that could be compared between different catchments.*

"Higher MBsum..." I would delete that because "higher" means more positive and here you mean more negative (it is always easier to talk about "more negative" or "more positive" glacier mass balances in order to avoid misunderstandings when using "higher" or "lower" mass balances!), moreover you use the abbreviations MBsum and MBwin but don't explain them (I can easily guess what it means but someone else not too familiar with glaciers might not at first glance); please see also my comments above about how you calculated "regional mass balances", I think if you calculate catchment-wide mass balance or mass balance anomalies in a more appropriate/more correct way you can use these numbers much better for a more realistic interpretation of "C" with the help of mass balance data...
*>> This part is removed from the table*

Table 3: Table caption: "in brackets" "in parentheses"; Add full stop at the end of the table caption; "Years most events" "Years with most events"; in my opinion it would be good to add the actual observation periods (e.g.19XY-19XY) in parentheses after "Alps", "Norway" and "Canada"!

*>> We made the changes but do not add the bservation periods because they are different for each catchment.*

**Review #2**

I found this study interesting and well-written. The findings are novel and the methodology is clear. My one concern is with the portion of the study dealing with glacier mass balance – which has already been brought up by in the review by Mauro Fischer, see my further general comment on this topic below.

The detailed review by Mauro Fischer caught most of the minor and technical comments that I would have included in my review. To save the authors time in responding to duplicate comments, I will not repeat them here. The few comments below are those I have tried to prune for overlap with Mauro's review. I enjoyed reading this study and, in my opinion, it is worthy of publication in HESS with some minor revision.

**General Comments**

Like the other reviewer, I have concerns about using the median of measured mass balance data. Using the area-weighted average, as has already been suggested, is a better option and has already been demonstrated in the response from the authors. I recognize the data limitations the authors are contending with, and I wonder if the mass balance analysis could just be removed from the study. I understand why the authors would like to include this type of analysis, but, in my opinion, it is a very minor part of the study and novelty and importance of the manuscript would not be hindered by removing this small piece. It would be a great avenue for future research. I will leave it up to the authors to decide whether a revised version of the mass balance analysis with the area-weighted averages should or should not be included in the revised manuscript.

*>> yes, we removed this part from the study.*

**Specific Comments**

*Discussion:*

Two sections in the discussion seem to overlap: '5.3 Drivers of Event-to-event variability in compensation levels', and '5.5 Temporal variability in event responses'. Both sections are discussing results presented section 4.4 (drivers of event-to-event variability). From the headings, it is not clear to me what the difference is between the two sections. I suggest combining the two sections into one.

*>> 5.3 and 5.5 were merged and part of 5.5 was removed.*

There are several places in the discussion where I found myself wanting a reference back to the relevant results. Below I've listed two locations. In my opinion, cohesion through the manuscript would be improved by adding a few figure, table, or section references to the discussion.

Line 381 – refer to the results supporting this stated finding.

Line 438-439 – refer to the results supporting the stated finding

*>> Thank you for pointing that out, we included more references to relevant results in the discussion.*

**Other minor comments:**

Figure 1 and Figure S1:  Are there just two line thicknesses used in these figures? What gc values do these thicknesses correspond to? What is the break value?

>>  No, the link thicknesses scale with the relative glacier cover. Since only a few catchments are highly glacierized, the differences between the lower glacierized catchments are minor. We added in the caption that line thickness corresponds to relative glacier cover – the thicker the line, the higher the relative glacier cover.

Figure 2 – Add a legend to clarify the relation between color and region.

>> We changed figure 1 instead of Figure 2?

Figure 8 – What are the glacier cover classes? Perhaps just list them in the caption.

>> They are in the topleft figure – this has been added to the caption

Table 3 – Heading should be 'Average duration of events [d]'?

>> Changed

Line 270: Missing comma in sentence starting with 'All variables,'

>> Changed

Line 393: 'and follow up studies Moyer et al. (2016)).' Seems like part of this sentence is missing

>> Changed